SPECIAL ISSUE
CELL BIOLOGY OF THE NUCLEUS

# CDK9 interacts with a RanGTP–importin-β complex to regulate erythroid enucleation

Lucas M. Newton[1,2,3,4,5], Krystle Y. B. Lim[1,2], Donia Y. Abeid[1,2], Christina B. Wölwer[6], Chad J. Johnson[7], Sarah M. Russell[3,4,5], Edwin D. Hawkins[8,9] and Patrick O. Humbert[1,2,10,11,*]

## ABSTRACT

Erythroid enucleation is the final stage of erythroid terminal differentiation and involves the separation of an orthochromatic erythroblast into two daughter cells – a pyrenocyte containing the extruded nucleus, and a reticulocyte, which will become a red blood cell. Our previous work has identified CDK9 as a regulator of erythroid enucleation that appears to act independently of its known role in regulating RNA polymerase II transcription, suggesting the potential for a new CDK role. Using a co-immunoprecipitation and mass spectrometry approach, we here identified the interactome of CDK9 in differentiating erythroblasts. We show that CDK9 interacts with a RanGTP–importin-β complex during erythroid terminal differentiation, and inhibition of importin-β in erythroblasts blocks erythroid enucleation. Using imaging analysis and functional assays of enucleating erythroblasts, we show that CDK9 and importin-β colocate at a crucial site of activity opposite to the nucleus before nuclear extrusion and we describe a novel finding that physically links CDK9 and importin-β activity prior to calmodulin and $Ca^{2+}$ signalling, and subsequent F-actin activity, to achieve enucleation.

KEY WORDS: Enucleation, Erythropoiesis, CDK9, Importin-β

## INTRODUCTION

Erythroid enucleation is the asymmetrical separation of the future erythrocyte from its nucleus and is the final stage of erythropoiesis before the developing erythroblast enters the peripheral blood stream (Palis, 2014). Enucleation is a rate limiting step in efforts to produce red blood cells *ex vivo* for transfusion medicine (Christaki et al., 2019; Gallego-Murillo et al., 2022), necessitating a clearer understanding of its regulation. Several recent reviews describe what is currently known of the molecular and cellular steps required for enucleation (Menon and Ghaffari, 2021; Moras et al., 2017; Newton et al., 2024). At a cellular level, chromatin compaction begins at the polychromatic erythroblast stage, one division prior to the orthochromatic erythroblast, which subsequently undergoes nuclear polarisation and nuclear extrusion; the anucleate cell, termed a reticulocyte, continues developing in the bone marrow for a short time while it undergoes further morphological changes and organelle clearance before moving into the circulation (Menon and Ghaffari, 2021; Moras et al., 2017; Newton et al., 2024). Importantly, the extruded nucleus, termed the pyrenocyte, is contained within a separate cell membrane and enclosed in a thin layer of cytoplasm, suggesting the mechanism is not purely exocytosis of nuclear material (Yoshida et al., 2005). Enucleation has been compared to both cytokinesis, and the apoptosis-driven denucleation events observed in lens epithelia and keratinocytes (Rogerson et al., 2018; Wride, 2011); however, the mechanisms that underpin enucleation appear quite distinct from either process (Newton et al., 2024). The molecular regulation of this unique cellular process is poorly understood, but we recently made the surprising finding that cyclin-dependent kinase-9 (CDK9) is essential for erythroid enucleation (Wölwer et al., 2015). Interestingly, our findings also suggested that CDK9 does not act through the canonical RNA Pol II-dependent pathway (Anshabo et al., 2021; Paparidis et al., 2017; Wölwer et al., 2015). Despite its importance, the precise mechanisms that govern enucleation remain relatively poorly understood.

Cytoskeletal elements such as F-actin and non-muscle myosin IIB (NMIIB) are required for the nuclear extrusion process, and although the formation of a contractile actomyosin ring (CAR) has been proposed (Ji et al., 2008; Konstantinidis et al., 2012; Koury et al., 1989; Li et al., 2017), other models suggest that enucleation is achieved primarily through the activity of an actin-mediated enucleosome and by membrane reorganisation by trafficking of endocytic vesicles (An et al., 2021; Keerthivasan et al., 2010; Liang et al., 2015; Nowak et al., 2017). $Ca^{2+}$ signalling through the calmodulin (CaM) pathway is crucial to achieve enucleation following uptake of extracellular $Ca^{2+}$ prior to nuclear extrusion (Wölwer et al., 2016). This is likely due to the ability for calmodulin kinase II (CaMKII) to modulate the actin cytoskeleton by altering F-actin cross-linking and the rate of F-actin polymerisation (Hoffman et al., 2013; Ohta et al., 1986; Sanabria et al., 2009).

Although many of the above factors are well established, the use of standard genetic approaches to identify new regulators of enucleation can be problematic in terms of their interpretation. This is because of the possible indirect effects of genetically mediated perturbation, when using CRISPR or RNAi approaches on upstream

[1]Department of Biochemistry & Chemistry, La Trobe University, Melbourne, VIC 3073, Australia. [2]La Trobe Institute for Molecular Science, La Trobe University, Melbourne, VIC 3073, Australia. [3]Optical Sciences Centre, Swinburne University of Technology, Hawthorn, VIC 3072, Australia. [4]Division of Cancer Research, Peter MacCallum Cancer Centre, Parkville, VIC 3010, Australia. [5]Sir Peter MacCallum Department of Oncology, The University of Melbourne, Melbourne, VIC 3010, Australia. [6]Institute of Pathology, University Hospital Cologne, Cologne 50937, Germany. [7]La Trobe Bioimaging Platform, La Trobe University, Melbourne, VIC 3073, Australia. [8]The Walter and Eliza Hall Institute of Medical Research, Parkville, VIC 3010, Australia. [9]Department of Medical Biology, University of Melbourne, Parkville, VIC 3010, Australia. [10]Department of Biochemistry & Pharmacology, University of Melbourne, Parkville, VIC 3010, Australia. [11]Department of Clinical Pathology, University of Melbourne, Parkville, VIC 3010, Australia.

*Author for correspondence (p.humbert@latrobe.edu.au)

L.M.N., 0000-0001-5984-5778; K.Y.B.L., 0000-0002-4008-6529; D.Y.A., 0009-0005-0686-0495; C.B.W., 0000-0003-1190-2408; S.M.R., 0000-0001-5826-9641; E.D.H., 0000-0002-3686-8261; P.O.H., 0000-0002-1366-6691

processes in terminal differentiation, which can indirectly result in reduced downstream enucleation. To circumvent this, we have developed and used a chemical genetics approach to identify novel regulators of enucleation. We initially identified CDK9 as a novel regulator of erythroid enucleation (Wölwer et al., 2015). CDK9 is predominantly known for its role as the catalytic subunit of the positive-transcription and elongation factor-B (P-TEFb) complex, which alongside its cyclin partner cyclin T1 (CycT1; encoded by *CCNT1*), cyclin T2a or b (CycT2a or b, encoded by *CCNT2*) or cyclin K (encoded by *CCNK*), plays a fundamental role in the regulation of RNA polymerase II (RNA Pol II) transcriptional activity (Paparidis et al., 2017; Gressel et al., 2017). However, our previous study has shown that blocking the activity of RNA Pol II in late erythropoiesis does not arrest enucleation (Wölwer et al., 2015), suggesting a new role for CDK9 in enucleation that is independent of RNA Pol II.

Here, using mouse and human erythroid culture models, we characterised the role of CDK9 in erythroid enucleation. We conducted imaging analysis of enucleating erythroblasts and found that CDK9 and cyclin T1 become more cytosolic and localised nearby F-actin prior to nuclear extrusion. Additionally, we showed that CDK9 localisation was dependent on F-actin activity. To identify the pathways CDK9 might be regulating in enucleation, we carried out a proteomics screen to identify relevant CDK9 interactors during erythroid enucleation. We showed that CDK9 associates with a RanGTP–importin-β complex, and that CDK9 and importin-β function upstream of $Ca^{2+}$ signalling and F-actin activity prior to nuclear extrusion. Our study provides new insights into the function of CDK9 in enucleation, and identifies a key link between the new regulator of enucleation importin-β and the previously described regulator CDK9 (Wölwer et al., 2015). As CDK9 inhibitors are being tested for clinical use against a variety of cancers, our study also highlights potential side-effects of CDK9 inhibitors on erythropoiesis and enucleation.

## RESULTS

### Phosphorylated CDK9 and cyclin T1 become cytoplasmic and localise near F-actin during nuclear extrusion

To better understand how CDK9 might regulate enucleation, we investigated the localisation of CDK9 and its key binding partner cyclin T1 during erythroid differentiation and enucleation. We used mouse and human erythroid *in vitro* systems to compare, for two stages of enucleation (Fig. 1A), the localisation and phosphorylation (at Thr186, to reflect CDK9 activation, denoted p-CDK9) of CDK9. In mouse orthochromatic erythroblasts p-CDK9 was observed in smaller clusters within the nucleus during the nuclear condensation phase, before localising to larger clusters within the cytoplasm that were also found in the future reticulocyte during extrusion; these clusters also seemed to localise closely with F-actin during nuclear extrusion (Fig. 1B). The p-CDK9 and CDK9 signal could be seen to colocalise; however, total CDK9 protein was more spread throughout the reticulocyte (Fig. 1B). Cyclin T1, the main binding partner and regulator of CDK9, was localised to the cytoplasm of mouse orthochromatic cells during nuclear condensation (Fig. 1C).

Upon measuring nuclear to cytoplasmic fluorescence ratios, we observed a strong shift towards the cytoplasm for p-CDK9 and total CDK9 (Fig. 1D). Cyclin T1 remained primarily cytosolic before colocalising with p-CDK9 in clustered regions near F-actin during nuclear extrusion (Fig. 1C,D). These findings were recapitulated during enucleation in the human erythroblast cell line HUDEP-2 (Fig. 1E,F). Quantifying these images revealed a complete exclusion of p-CDK9, CDK9 and cyclin T1 in day 12 (D12) differentiated HUDEP-2 cells from the nucleus, which appeared highly condensed

(Fig. 1G). Thus, CDK9 remains active in the nucleolus very late into terminal differentiation and subsequently localises with F-actin and cyclin T at the rear of the extruding nucleus before relocating to the future reticulocyte in both mouse and human enucleating erythroblasts.

### Highly selective inhibition of CDK9 activity arrests enucleation at the same stage as F-actin induced enucleation block

Our previous study has revealed that pharmacological blockage of CDK9 activity leads to an arrest in enucleation, and together with the observation that CDK9–cyclin T1 colocalise with F-actin, suggests that CDK9 might regulate the extrusion event (Wölwer et al., 2015). To examine this potential functional significance, we initially compared the effects of CDK9 inhibition with inhibition of actin polymerisation. With the advent of next-generation high-selectivity CDK9 inhibitors, we first confirmed the effect of specific CDK9 inhibition on enucleating cells in comparison to the F-actin inhibitor cytochalasin D (CytoD), which consistently arrests cells at a late stage of enucleation (Konstantinidis et al., 2012; Wang et al., 2012; Wölwer et al., 2015) (Fig. 2). Treatment of orthochromatic mouse erythroblasts (Fig. 2A) with CDK9 inhibitors NVP-2 (Olson et al., 2017) (Fig. 2B) and AZD4573 (Barlaam et al., 2020) (Fig. 2C) arrested enucleation in a dose-dependent manner, with a mean reduction of 51% and 64% against the vehicle (DMSO) control at 10 nM, respectively. Phenotype analysis of cytospun cells revealed similar arrest patterns for both inhibitors, which is consistent with the phenotypes reported previously for less-specific CDK9 inhibitors (Wölwer et al., 2015), with an increased percentage of cells with a polarised or partially extruded nucleus (Fig. 2B,C) compared to the control (Fig. 2A). NVP-2 also reduced the enucleation of human cells compared to the vehicle (DMSO) control, with a similar increase in polarised nuclei (Fig. S1A,B). These data confirm that the action of CDK9 on erythroid enucleation is conserved between mouse and human, and further validates our previous findings by alleviating concerns around inhibitor off-target effects on other CDKs described previously (Wölwer et al., 2015).

All other methods of chemical inhibition of CDK9 provided similar results. Atuveiclib (BAY 1143572) (Lücking et al., 2017) and the thalidomide-conjugate PROTAC degrader THAL-SNS-032 (Olson et al., 2017) both arrested enucleation compared to the vehicle (DMSO) control (Fig. S2). To compare the effects of CDK9 inhibition to a CytoD-induced late-stage enucleation blockage, we treated mouse orthochromatic erythroblasts with a range of concentrations and quantified the phenotype of cytospins using morphology indicators (Fig. 2). Examination of cytospins indicated that treatment with CDK9 inhibitors phenocopied the effect of a blockage by the actin polymerisation inhibitor CytoD (Fig. 2D), with quantitation of morphology indicating that CytoD caused a slightly later arrest compared to CDK9 inhibitors, with more cells showing highly polarised and bulging nuclei (Fig. 2D). We conclude that CDK9 must act just prior to the final stages of enucleation, with the observed blockage closely resembling that caused by F-actin inhibition.

### CDK9 is active upstream of $Ca^{2+}$ signalling and F-actin polymerisation prior to nuclear extrusion

Our localisation studies showed CDK9 colocalisation with F-actin at late stages of enucleation and suggest that CDK9 is required just upstream of F-actin and therefore might play a role in regulating the nuclear extrusion event. To confirm the order in which activated CDK9 and F-actin assembly might be required for enucleation,

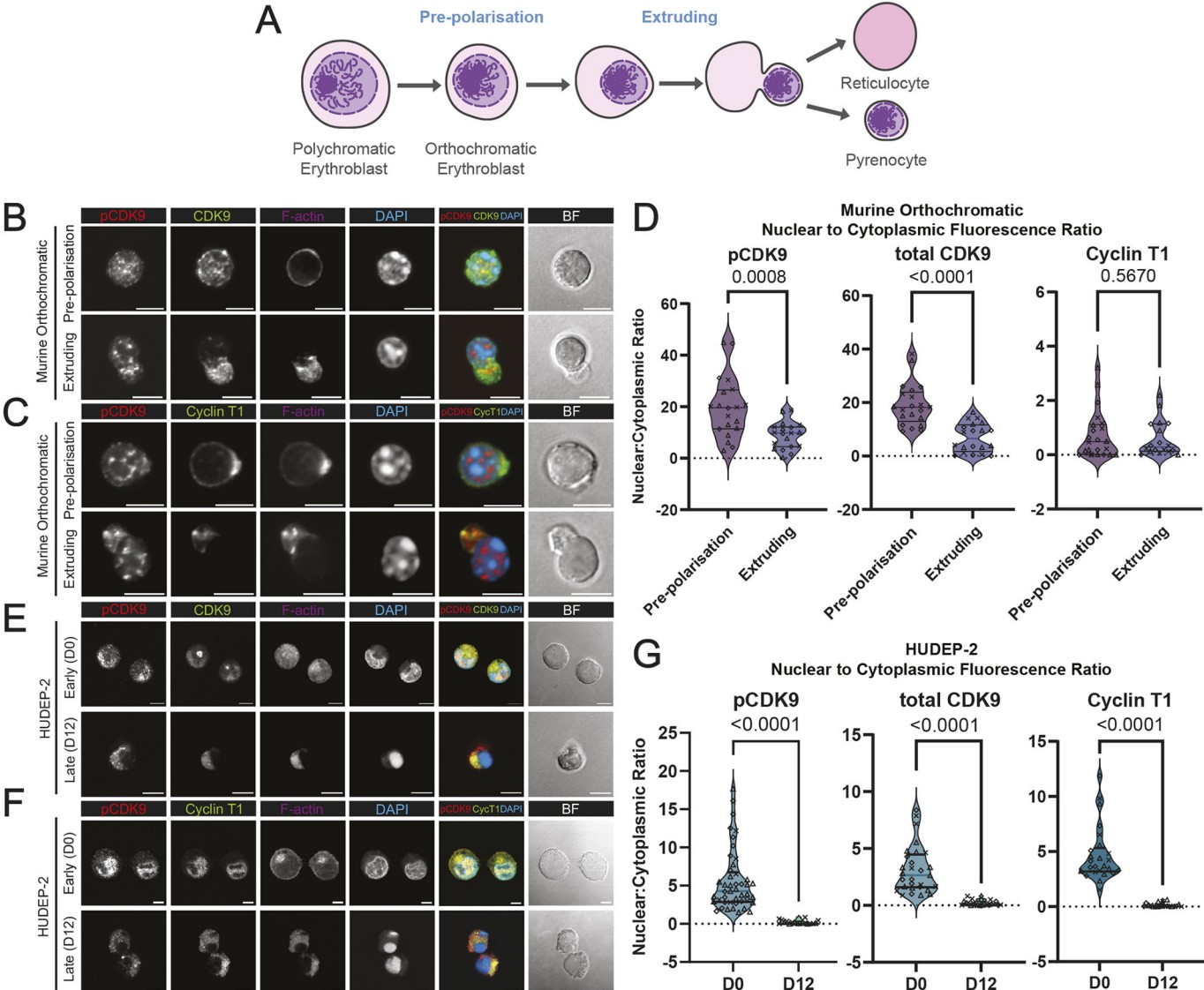

**Fig. 1. Phosphorylated CDK9 at Thr186 and cyclin T1 are excluded from the nucleus during nuclear extrusion and localise to the cytoplasm and future reticulocyte following enucleation.** (A) Diagram depicting the stages of erythroid enucleation described in this study. (B) Immunofluorescence confocal microscopy of mouse orthochromatic erythroblasts at the pre-polarisation and extrusion phases of erythroid enucleation stained for phospho-CDK9(Thr186), CDK9 (F-6), phalloidin for F-actin and DAPI for nuclei. Brightfield (BF) images and actin are excluded from the merge. Yellow colour on the merge indicates colocalisation of p-CDK9 and CDK9. (C) Immunofluorescence confocal microscopy of mouse orthochromatic erythroblasts at the pre-polarisation and extrusion phases of erythroid enucleation stained for phospho-CDK9(Thr186), Cyclin T1, phalloidin for F-actin and DAPI for nuclei. Yellow colour on the merge indicates colocalisation of p-CDK9 and cyclin T1. (D) Nuclear to cytoplasmic ratios of p-CDK9 ($n$=20 pre-polarisation, 19 extruding orthoblasts), total CDK9 ($n$=21 pre-polarisation, 20 extruding orthoblasts) and cyclin T1 ($n$=22 pre-polarisation, 18 extruding orthoblasts). (E) Immunofluorescence confocal microscopy of HUDEP-2 cells at early (day 0) and late (day 12) of differentiation stained for phospho-CDK9(Thr186), CDK9 (F-6), phalloidin for F-actin and DAPI for nuclei. Yellow colour on the merge indicates colocalisation of p-CDK9 and CDK9. (F) Immunofluorescence confocal microscopy of HUDEP-2 cells at early (day 0) and late (day 12) of differentiation stained for phospho-CDK9(Thr186), cyclin T1, phalloidin for F-actin and DAPI for nuclei. Yellow colour on the merge indicates colocalisation of p-CDK9 and cyclin T1. (G) Nuclear to cytoplasmic ratios of p-CDK9 [$n$=45 day 0 (D0), 18 day 12 (D12) HUDEP-2 cells], total CDK9 [$n$=29 day 0 (D0), 22 day 12 (D12) HUDEP-2 cells] and cyclin T1 [$n$=25 day 0 (D0), 19 day 12 (D12) HUDEP-2 cells]. In B,C,E,F, brightfield (BF) images and actin are excluded from the merge. Lines in violin plots in D and F highlight median and quartiles. Scale bars: 5 µm.

we first examined the reciprocal localisation of activated CDK9 and F-actin in the presence of NVP-2 and CytoD in mouse orthochromatic erythroblasts using immunofluorescence imaging (Fig. 3A). CytoD, which consistently arrests the cells with a partially extruded nucleus, resulted in accumulation of p-CDK9 at the site of enucleation (Fig. 3A). In addition, the p-CDK9 signal was diminished in cells treated with NVP-2, as might be expected (Fig. 3A). F-actin appeared to accumulate and colocalise with p-CDK9 in CytoD-treated cells but appeared to be dispersed near

the cell membrane in NVP-2-treated cells, where F-actin would normally localise toward the site of enucleation (Fig. 3A). Although these experiments indicate a close relationship between CDK9 and F-actin in the context of enucleation, they do not clarify the signalling hierarchy between CDK9 and F-actin.

$Ca^{2+}$ signalling is required for effective enucleation, but the positioning of $Ca^{2+}$ signalling components is not known (Wölwer et al., 2016). We found that in orthochromatic erythroblasts, CaMKII, a key component in CaM and $Ca^{2+}$ signalling, localised primarily in

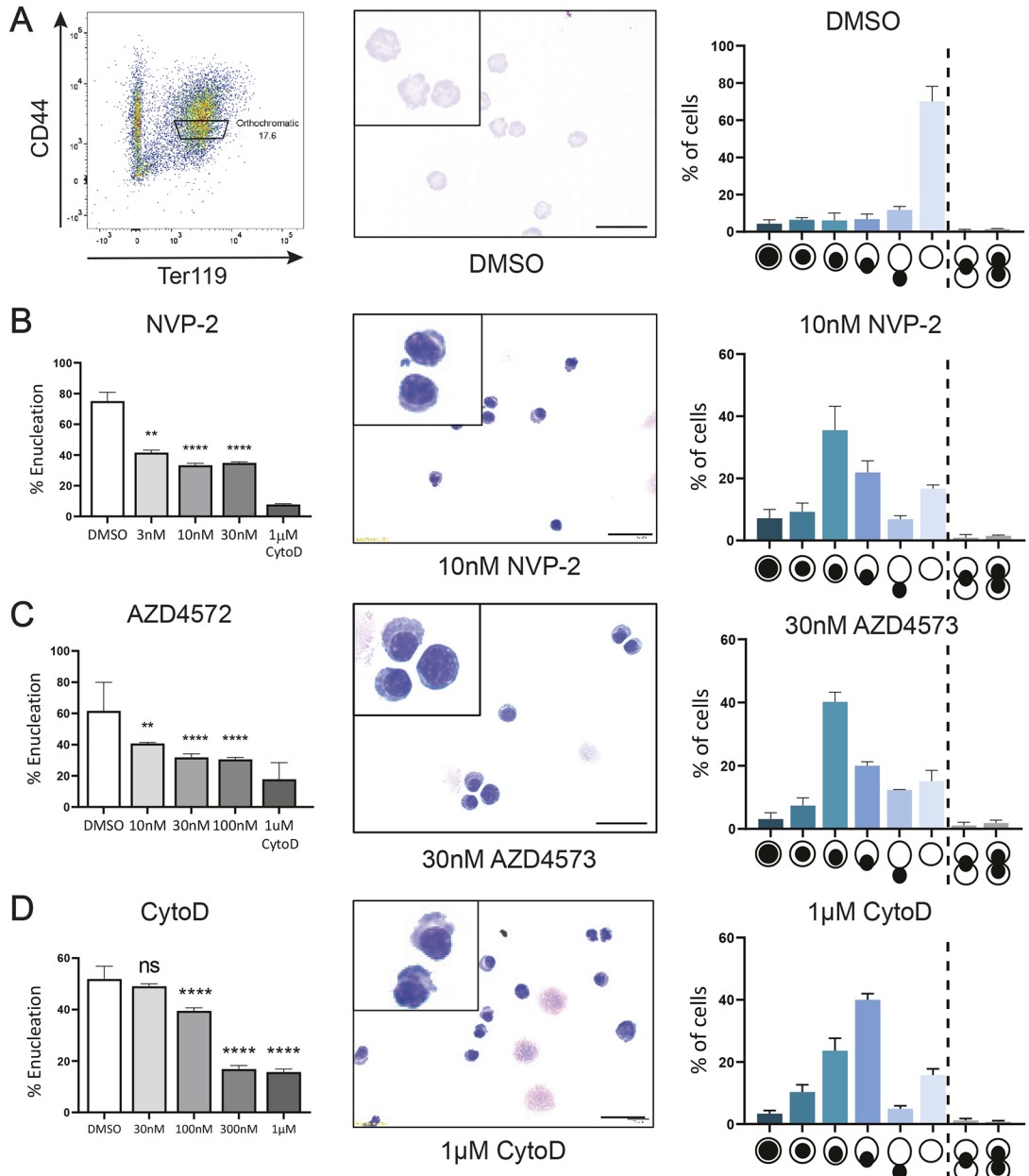

**Fig. 2. CDK9 inhibitors cause an enucleation arrest that phenocopies CytoD-mediated enucleation blockage.** (A) Gating strategy used for FACS isolation of orthochromatic erythroblasts from mouse spleen following phenylhydrazine treatment alongside cytospin Rapid Diff staining and phenotype analysis following 12-h incubation of isolated cells. (B) Quantification of enucleation of mouse orthochromatic erythroblasts following treatment with NVP-2 for 12 h. DMSO (vehicle control) and CytoD (positive control) are included, in addition to cytospin Rapid Diff staining and phenotype analysis. $n$=4 replicates across three independent experiments. Phenotype symbols refer the position of the nucleus (black circle) within the enucleating erythroblast (white circle). **$P$<0.01, ****$P$<0.0001 (one-way ANOVA with Dunnett's multiple comparisons test). (C) Quantification of enucleation of mouse orthochromatic erythroblasts following treatment with AZD4573 for 12 h. DMSO (vehicle control) and CytoD (positive control) are included, in addition to cytospin Rapid Diff staining and phenotype analysis. $n$=4 replicates across three independent experiments. **$P$<0.01, ****$P$<0.0001 (one-way ANOVA with Dunnett's multiple comparisons test). (D) Quantification of enucleation of mouse orthochromatic erythroblasts following treatment with CytoD for 12 h. DMSO (vehicle control) is included, in addition to cytospin Rapid Diff staining and phenotype analysis. $n$=4 replicates across two independent experiments. ns, not significant, ****$P$<0.0001; one-way ANOVA with Dunnett's multiple comparisons test). All error bars show mean±s.e.m. Scale bars: 10 μm.

the cytoplasm during nuclear polarisation, and colocalised with CDK9 at the rear of the nucleus during nuclear extrusion (Fig. 3B). To better observe the relationship between CDK9 and CaMKII, we examined their localisation in orthochromatic erythroblasts treated with either 100 nM NVP-2 or 5 μM KN-62, which inhibits CaMKII activity (Okazaki et al., 1994) (Fig. 3C). CDK9 and CaMKII appeared to strongly colocalise under NVP-2- and KN-62-induced blockage, possibly suggesting the existence of a checkpoint

dependent on the activity of CDK9 and/or CaMKII. However, we again observed an accumulation of CDK9 near the membrane opposite the polarised nucleus under KN-62 blockage (Fig. 3C), suggesting that CaMKII signalling might be dependent on upstream CDK9 signalling to facilitate downstream F-actin-mediated nuclear extrusion. We quantified these accumulations of p-CDK9, which we term cytosolic p-CDK9 puncta, which appeared in almost all CytoD-treated cells and in some cells treated with KN-62, but never under

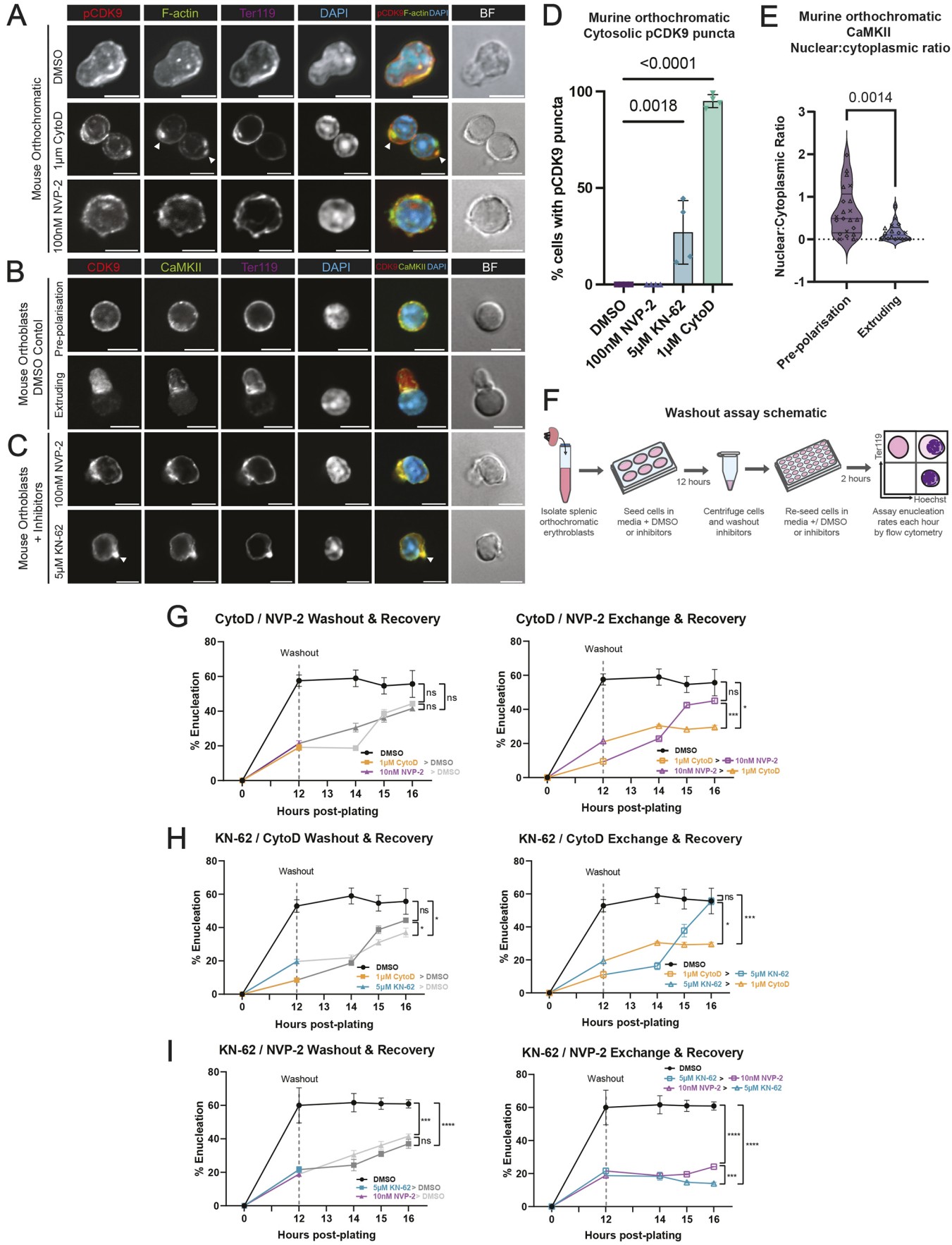

**Fig. 3.** See next page for legend.

**Fig. 3. F-actin is required downstream of CDK9 to achieve enucleation, and blocking enucleation with CytoD causes a build-up of active CDK9 that colocalises with F-actin.** (A) Immunofluorescence confocal microscopy of mouse orthochromatic erythroblasts treated with DMSO (vehicle control), 1 µM CytoD or 100 nM NVP-2 for 12 h and stained for phospho-CDK9(Thr186) phalloidin for F-actin, Ter119 and DAPI for nuclei. Yellow colour on the merge indicates colocalisation of p-CDK9 and F-actin. (B) Immunofluorescence confocal microscopy of mouse orthochromatic erythroblasts at the pre-polarisation and extrusion phases of erythroid enucleation treated with DMSO (vehicle control) and stained for CDK9 (F-6), CaMKII, Ter119 and DAPI for nuclei. (C) Immunofluorescence confocal microscopy of mouse orthochromatic erythroblasts following 12 h treatment with 100 nM NVP-2 or 5 µM KN-62, stained for CDK9 (F-6), CaMKII, Ter119 and DAPI for nuclei. In A–C, brightfield (BF) and Ter119 are excluded from the merge. Arrowheads indicate accumulation of CDK9. (D) Manual scoring of p-CDK9 puncta in orthoblasts following treatment with DMSO (vehicle; $n$=22 cells averaged across four experiments), 1 µM CytoD ($n$=27 cells averaged across four experiments), 100 nM NVP-2 ($n$=21 cells averaged across four experiments) and 5 µM KN-62 ($n$=18 cells averaged across four experiments). (E) Nuclear-to-cytoplasmic ratio of CaMKII ($n$=21 pre-polarisation, 18 extruding orthoblasts). (F) Schematic diagram describing workflow for inhibitor washout and exchange functional assays used in this study. (G) Enucleation rates of mouse orthochromatic erythroblasts following inhibitor washout into fresh medium containing DMSO (left) or inhibitors (CytoD or NVP-2; right) at 12 h post-plating. Enucleation rates were measured at 12, 14, 15 and 16 h post-plating. DMSO (black; negative control) is included. Enucleation rates at hour 16 were statistically compared. (H) Enucleation rates of mouse orthochromatic erythroblasts following inhibitor washout into fresh media containing DMSO (first plot) or inhibitors (CytoD or KN-62) at 12 h post-plating. Enucleation rates were measured at 12, 14, 15 and 16 h post-plating. DMSO (black; negative control) is included. Enucleation rates at hour 16 were statistically compared. (I) Enucleation rates of mouse orthochromatic erythroblasts following inhibitor washout into fresh media containing DMSO (first plot) or inhibitors (KN-62 or NVP-2) at 12 h post-plating. Enucleation rates were measured at 12-, 14-, 15- and 16-h post-plating. DMSO (black; negative control) is included. Enucleation rates at hour 16 were statistically compared. For G–I, $n$=5 replicates across two independent experiments. ns, not significant; *$P$<0.05; ***$P$<0.001; ****$P$<0.0001 (two-way ANOVA with Tukey's multiple comparisons test). In G–I, colours change to match either DMSO (grey) or inhibitors (coloured) after washouts. All error bars show mean±s.e.m. Lines in violin plots in E highlight median and quartiles. Scale bars: 5 µm.

NVP-2 or DMSO treatment (Fig. 3D). Additionally, assessment of nuclear-to-cytoplasmic ratio of CaMKII revealed a significant shift to the cytoplasm in extruding orthochromatic erythroblasts (Fig. 3E).

To assess the order in which CDK9, actin and CaMKII act during enucleation, we applied reversible inhibitors in sequence (Fig. 3G–I). We reasoned that the kinetics of recovery would differ depending on whether downstream events were inhibited before or after upstream events. If acting before, they should not affect recovery, but if acting after, recovery would be slower or absent. To ensure minimal toxicity, we assessed the viability of primary mouse orthochromatic erythroblasts treated with the inhibitors throughout this study using propidium iodide (PI) staining and found that none reduced overall viability by more than 5% at the chosen concentrations (Fig. S3). Importantly, PI-positive cells were excluded from all flow cytometry analysis. NVP-2 acts to inhibit CDK9 by reversibly blocking the ATP-binding site (Olson et al., 2017), and CytoD reversibly disrupts actin polymerisation by forcing actin dimerisation (Friederich et al., 1993; Goddette and Frieden, 1986; Stevenson and Begg, 1994). Additionally, KN-62 can reversibly inhibit CaMKII (Okazaki et al., 1994). As NVP-2, CytoD and KN-62 are reversible inhibitors, we used these inhibitors to study the order of events prior to nuclear extrusion. A schematic describing the workflow of these experiments is provided (Fig. 3F). Enucleation rates were assessed by flow cytometry following 12 h incubation with 1 µM CytoD or 10 nM

NVP-2 and compared to the vehicle (DMSO) control directly after washout, and at hour 14, 15 and 16 after plating cells (Fig. 3G). Enucleation rates following washout of CytoD or NVP-2 into medium containing DMSO carrier were shown to recover close to the rate of the DMSO vehicle control at hour 16, confirming the reversibility of both competitive inhibitors (Fig. 3G, left). Exchanging CytoD for NVP-2 at the washout point resulted in a recovery of enucleation like the control, however, exchanging NVP-2 for CytoD resulted in impaired recovery of enucleation to a statistically significant degree compared to both the control and the reverse inhibitor exchange at hour 16 (Fig. 3G, right). These experiments indicate that cells arrested during F-actin polymerisation inhibition by CytoD are not affected by later CDK9 inhibition. Together these results indicate that CDK9 acts upstream of F-actin for nuclear extrusion.

Next, we examined the relationship between actin and CDK9 with CaMKII. Enucleation rates recovered following washout of CytoD and KN-62, with CytoD-treated cells recovering with non-significant variance to the control, and KN-62 washout cells partially recovering, but with enucleation rates still significantly lower than the control at hour 16 (Fig. 3H, left). Inhibition of F-actin (CytoD) after CaMKII signalling (KN-62) resultd in a complete recovery of enucleation rates by hour 16 compared to the control, but the reverse sequence did not (Fig. 3H, right). This indicates that CaMKII activity is required upstream of F-actin polymerisation prior to nuclear extrusion. Similarly, enucleation rates were assessed following 12 h incubation with KN-62 or NVP-2 and compared to the vehicle (DMSO) control, where enucleation rates recovered equally in both the KN-62 and NVP-2 treatment to non-significantly different levels compared to the control at hour 16 (Fig. 3I, left). Although washout of KN-62 into NVP-2 allowed for a slight recovery of enucleation, it appeared that no sequence could be conferred, which might suggest that CDK9 and CaM signalling occurs concurrently or independently (Fig. 3I, right). Taken together, these results confirm that both CDK9 activity and CaM signalling is required before F-actin polymerisation during enucleation.

## CDK9 is associated with a Ran–NEMP1–importin-β complex in terminally differentiating erythroid cells

Having established that CDK9 acts closely with $Ca^{2+}$ signalling and is upstream of F-actin in its regulation of enucleation, and that our previous work shows that the typical downstream effector of CDK9, namely RNA Pol II, is dispensable for enucleation (Wölwer et al., 2015), we assessed how CDK9 might modulate actin activity and enucleation in general. We undertook a comprehensive proteomics screen to identify new interactors of CDK9 in the context of erythroid differentiation using CDK9 co-immunoprecipitation mass spectrometry (co-IP-MS). We used the human HUDEP-2 cell line, as this provided both the sufficient material and the ability to manipulate the system genetically. First, we investigated the interactome of endogenous CDK9 in undifferentiated (day 0) and differentiated (day 6) HUDEP-2 cells. We used day 6 differentiated HUDEP-2 cells due to better cell viability and overall protein availability compared to more highly differentiated cells. The original protein database search is available in Table S2, however, for our analysis we took a stringent cut-off approach, including proteins that were identified in all 3 replicates and ≥3 fold enriched compared to the IgG co-IP controls. 65 unique proteins were identified in undifferentiated HUDEP-2 cells, and 114 unique proteins identified in differentiated HUDEP-2 cells, with 13 proteins identified in both data sets (Fig. 4A). We performed a STRING interactome analysis on these 13 proteins. Of note, we identified CDK9 and cyclin T1

(CCNT1) as common to both data sets, validating the presence of a CDK9–cyclin T1 complex in differentiating human erythroid cells (Fig. 4B). Other known interactors of CDK9, including vimentin (VIM) and the DNA-activated protein kinase subunit PRKDC were also found in both datasets (Yang et al., 2015; Zhang et al., 2014). Surprisingly, we also found five members of the Ran GTPase nuclear import complex, including nuclear envelope membrane protein 1 (NEMP1, also known as TMEM194A) and importin-β (KPNB1) (Fig. 4B). Of note, NEMP1 has recently been described as being an essential regulator of enucleation in the mouse (Hodzic et al., 2022), confirming our proteomics screen using human cells can identify functionally relevant proteins for enucleation.

We next explored whether CDK9 activity impacted the interactome by comparing this upon expression of exogenous CDK9, the constitutively active kinase mutant CDK9-T186A

and the dominant-negative kinase mutant CDK9-D168N (Dow et al., 2010). We isolated a polyclonal population of transduced cells by fluorescence-activated cell sorting (FACS) (Fig. S4) and subsequently performed CDK9 co-IP-MS in day 6 differentiated cells. With the cut-off described above, we identified a total of 243 proteins, with 98 uniquely identified in the CDK9 expression line, 65 of which were unique to the CDK9-T186A line and 28 unique to the CDK9-D168N line, with nine proteins identified in all three sets (Fig. 4C). A STRING analysis of these nine shared proteins identified again three members of the Ran GTPase nuclear import complex, including importin-β, confirming our previous results (Fig. 4D). We then performed STRING interactome analysis on all 243 identified proteins and clustered the results using k-means clustering to understand the variety of processes and pathways that CDK9 might regulate in differentiating human erythroblasts

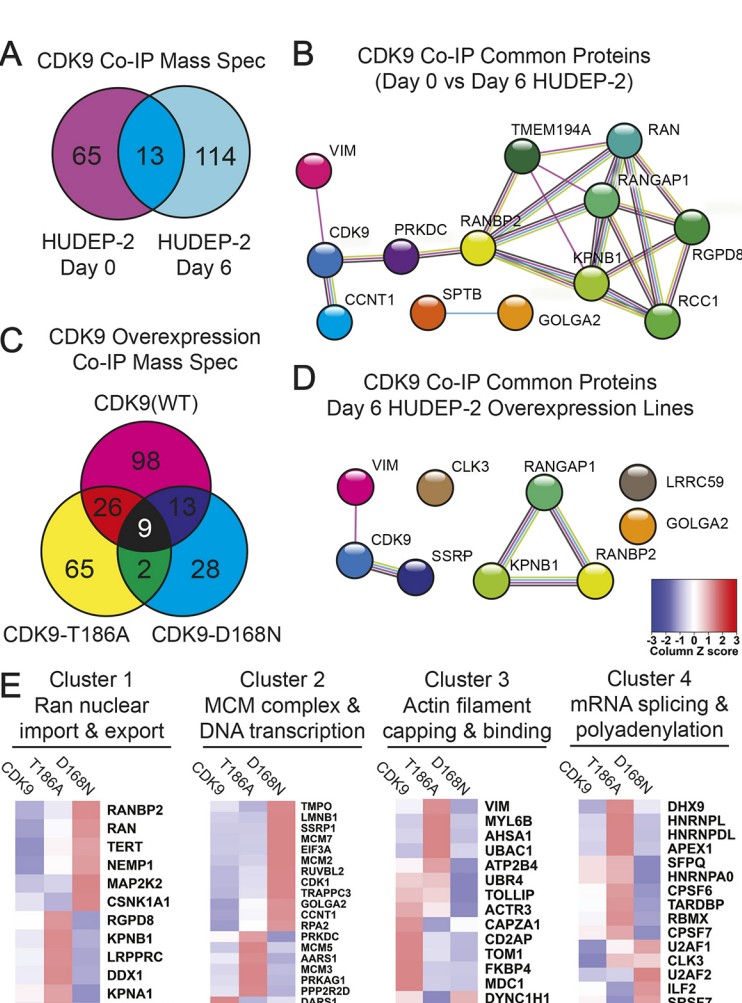

**Fig. 4. Summary of CDK9 co-immunoprecipitation mass spectrometry results in HUDEP-2 cells.** (A) Venn diagram of identified interactors of CDK9 in undifferentiated (Day 0) and differentiated (Day 6) HUDEP-2 cells. Proteins were considered an interactor when present in all three replicates and ≥3 fold enriched compared to the IgG controls. See Table S2 for complete list of identified proteins. (B) STRING analysis of the 13 common proteins identified in both undifferentiated (Day 0) and differentiated (Day 6) HUDEP-2 cells. STRING analysis was performed using medium confidence (0.4) for interaction score. Aqua lines represent known interactions from curated databases; purple lines represent experimentally determined known interactors. Green, red and blue lines represent predicted interactors; yellow lines represent text mining identification and black lines represent known co-expression. (C) Venn diagram of identified interactors of CDK9 overexpression, CDK9-T186A and CDK9-D168N lines in differentiated (Day 6) HUDEP-2 cells. Proteins were considered an interactor when present in all three replicates and ≥3 fold enriched compared to the IgG controls. See Table S2 for a complete list of identified proteins. (D) STRING analysis of the nine common proteins identified in all three HUDEP-2 CDK9 overexpression cell lines. STRING analysis was performed using medium confidence (0.4) for interaction score. Aqua lines represent known interactions from curated databases; purple lines represent experimentally determined known interactors. Green, red and blue lines represent predicted interactors; yellow lines represent text mining identification and black lines represent known co-expression. (E) Heatmaps of clusters 1–4 described in Fig. S5. Proteins identified in each cluster were compared to the average abundance found in CDK9, CDK9-T186A and CDK9-D168N overexpression HUDEP-2 cell lines. Top biological processes (GO) and molecular functions (GO) identified in STRING are listed below for each cluster (strength >2, false discovery P<0.001).

(Fig. S5). 11 clusters included processes related to Ran-mediated nuclear export and import (cluster 1), minichromosomal maintenance complex (MCM) and DNA transcription (cluster 2), actin filament capping and binding (cluster 3), and mRNA splicing and polyadenylation (cluster 4). These four clusters were selected for further analysis based on overall higher abundance in mass spectrometry protein identification (Fig. 4E). The abundance of proteins in each cluster was compared across the CDK9, CDK9-T186A and CDK9-D168N datasets (Fig. 4E). Within the nuclear trafficking cluster (cluster 1), importin-β (KPNB1), which drives nuclear import, was enriched in the constitutively active kinase mutant (T186A), but exportin-1, which drives nuclear export, was depleted (Fig. 4E, cluster 1). This is compatible with the notion that CDK9 phosphorylation might regulate its nucleocytoplasmic shuttling, as has been previously reported (Napolitano et al., 2002).

CDK9 and cyclin T1 were more abundant in the dominant-negative (D168N) CDK9 mutant lines, indicative of a relationship between CDK9 phosphorylation state and turnover of CDK9 and cyclin T1 (Fig. 4E, cluster 2). Additionally, CDK9, cyclin T1 and other interactors within the minichromosomal maintenance (MCM) complex and DNA transcription cluster (cluster 2) were enriched in the dominant-negative kinase mutant (D168N), suggesting that CDK9 kinase activity might impact the expression of various components involved in DNA transcription, including cyclin T1 and CDK9 itself (Fig. 4E, cluster 2). Interestingly, CDK9 has been shown to regulate DNA damage responses through MCM complex members through interaction with cyclin K (Yu and Cortez, 2011), which was not detected in our assay. CDK9 was found to interact with several components involved in actin filament capping and binding, which might have implications for how CDK9 affects actin-related processes during enucleation (Fig. 4E, cluster 3). Finally, examination of the mRNA splicing and polyadenylation complex reveals shifts in how CDK9 kinase functionality might impact on its previously reported role in mRNA splicing (Fig. 4E, cluster 4) (Hu et al., 2021). Here, our identification of a physical association between CDK9 and the Ran–importin-β nuclear import complex ties together our original finding of an RNA Pol II-independent role for CDK9 and recent findings that NEMP1 regulates nuclear envelope openings during erythroid enucleation.

## Importin-β is required for erythroid enucleation upstream of Ca²⁺ signalling and F-actin polymerisation

We next tested the function of importin-β in the context of enucleation using the reversible inhibitor of importin-β-mediated nuclear import importazole (Soderholm et al., 2011). Inhibition of importin-β using importazole arrested enucleation in a dose-dependent manner in mouse orthochromatic erythroblasts, demonstrating that importin-β function is crucial for enucleation (Fig. 5A). NEMP1, which has been implicated in maintenance of nuclear envelope openings, which are essential for enucleation (Hodzic et al., 2022), was identified in complex with CDK9 and importin-β, and so we aimed to assess the localisation of importin-β during both normal and arrested enucleation as a comparison. We assessed the effect of various concentrations of importazole on orthochromatic erythroblasts and found a dosage of 10 µM to have the most reliable effect without significantly increased toxicity (Fig. S3), which is comparable to or lower than concentrations used in other *in vitro* experiments (Soderholm et al., 2011; Yoshino et al., 2021; Bird et al., 2013; Kublun et al., 2014). Morphological analysis of orthochromatic erythroblasts treated with importazole indicated an increased number of cells arrested with a polarised nucleus (Fig. 5A), phenocopying both CDK9 and F-actin inhibitors and suggesting that importin-β is acting prior to nuclear extrusion.

Additionally, we treated day 12 differentiated HUDEP-2 cells with importazole for 24 h and counted enucleated cells versus non-enucleated and found a significant and dose-dependent reduction in enucleation, indicating a conserved function for importin-β in human erythroid enucleation (Fig. 5B). To test whether importin-β regulated the nucleocytoplasmic import of CDK9, we imaged fixed undifferentiated HUDEP-2 cells following treatment with NVP-2, importazole or DMSO to assess the nuclear-to-cytoplasmic ratio of active CDK9, cyclin T1 and importin-β. Blocking CDK9 activity did not alter the nucleocytoplasmic ratio of CDK9, cyclin T1 or importin-β (Fig. S6). Interestingly, importazole significantly altered the localisation of cyclin T1, but not CDK9 (Fig. S6). This result, taken together with observed co-immunoprecipitation of CDK9 and importin-β (Fig. 4), suggests that importin-β might interact with CDK9 for reasons other than nucleocytoplasmic transport in the context of enucleation.

As importazole appeared to block enucleation at a similar stage to CDK9 and F-actin inhibitors, we assessed which of these proteins might regulate each other during enucleation. We first examined importin-β localisation using fluorescence confocal microscopy. Importin-β appeared to localise to the nuclear envelope in pre-polarised erythroblasts (Fig. 5C) and was observed in the cytoplasm and near folded regions at the nuclear envelope, before colocalising with p-CDK9 in the future reticulocyte during nuclear extrusion (Fig. 5C). Cells arrested by treatment with either CytoD or importazole showed an accumulation of p-CDK9 at the site of enucleation, with importin-β localising around the membrane opposite to the polarised nucleus (Fig. 5D). Because p-CDK9 accumulated to the extrusion site in the same way under both actin and importin-β inhibitors, we postulate that p-CDK9, in a similar manner to actin, might be required for enucleation upstream of importin-β before nuclear extrusion.

To test this functionally, we performed the previously described inhibitor washout, exchange and recovery enucleation assays to determine an order of events using importazole, CytoD and NVP-2. Enucleation rates were assessed by flow cytometry following 12 h incubation with importazole, CytoD or NVP-2 and compared to the vehicle (DMSO) control directly after washout, and at hour 14, 15 and 16 after plating cells (Fig. 5E,F). Enucleation rates in the vehicle (DMSO) treated control group remain steady, and enucleation rates following importazole washout were able to recover, confirming the reversibility of the blockage by importazole (Fig. 5E, left). Inhibition of importin-β (importazole) before F-actin (CytoD) led to an initial recovery of enucleation rates but an overall reduction by hour 16 (Fig. 5E, right). Conversely, blocking F-actin (CytoD) before importin-β (importazole) resulted in a gradual recovery of enucleation rates (Fig. 5E, right). This indicates that importin-β is required at an earlier stage of enucleation than F-actin polymerisation. Blocking importin-β (importazole) before CDK9 (NVP-2) resulted in a consistent recovery of enucleation rates, but the reverse sequence did not (Fig. 5F). We repeated this by inhibiting CaMKII (KN-62) and importin-β (importazole); however, we did observe altered recovery rates for KN-62 in this experiment, where recovery was significantly reduced compared to the control at hour 16 (Fig. 5G, left). Nonetheless, we could observe a clear shift in recovery when washing out importazole into KN-62, which resulted in a reduction of enucleation, as opposed to washout of KN-62 into importazole, which allowed for a partial recovery in enucleation (Fig. 5G, right). Taken together, this supports our previous findings that CaMKII acts upstream of F-actin and indicates that CaM signalling is necessary downstream of CDK9 but might occur dynamically or simultaneously to importin-β activity. We conclude that importin-β is essential for enucleation, acting downstream of CDK9 and upstream of F-actin, and

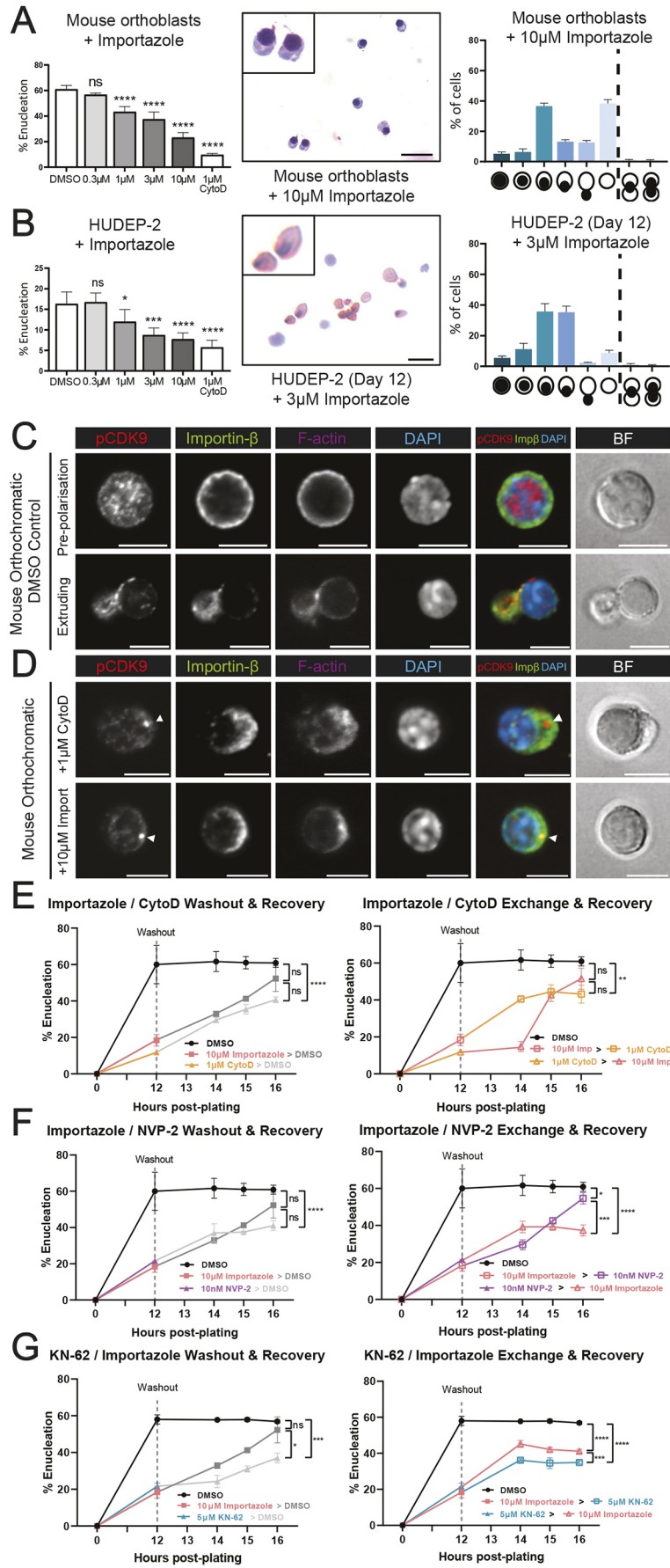

**Fig. 5. Importin-β activity is required for erythroid enucleation processes downstream of CDK9.** (A) Quantification of enucleation of mouse orthochromatic erythroblasts following treatment with importazole for 12 h. DMSO (vehicle control) and CytoD (positive control) are included, in addition to cytospin Rapid Diff staining and phenotype analysis. *n*=4 replicates across three independent experiments. ns, not significant; ****P<0.0001 (one-way ANOVA with Dunnett's multiple comparisons test). (B) Quantification of enucleation of day 12 differentiated HUDEP-2 cells following treatment with importazole for 12 h. DMSO (vehicle control) and CytoD (positive control) are included, in addition to cytospin Rapid Diff staining and phenotype analysis. *n*=4 replicates across three independent experiments. ns, not significant, *P<0.05, ***P<0.001, ****P<0.0001 (one-way ANOVA with Dunnett's multiple comparisons test). (C) Immunofluorescence confocal microscopy of DMSO (vehicle control)-treated mouse orthochromatic erythroblasts at the pre-polarisation and extrusion phases of erythroid enucleation stained for phospho-CDK9(Thr186), importin-β, phalloidin for F-actin and DAPI for nuclei. (D) Immunofluorescence confocal microscopy of mouse orthochromatic erythroblasts following 12 h treatment with 1 μM CytoD or 10 μM importazole, stained for phospho-CDK9(Thr186), importin-β, phalloidin for F-actin and DAPI for nuclei. Arrows indicate accumulation of CDK9. In C and D, brightfield (BF) and F-actin are excluded from the merge. Symbols in right-hand graphs are as per Fig. 2. (E) Enucleation rates of mouse orthochromatic erythroblasts following inhibitor washout into fresh medium containing DMSO (left) or inhibitors (CytoD or importazole; right) at 12 h post-plating. Enucleation rates were measured at 12, 14, 15 and 16 h post-plating. DMSO (black; negative control) is included. Enucleation rates at hour 16 were statistically compared. (F) Enucleation rates of mouse orthochromatic erythroblasts following inhibitor washout into fresh media containing DMSO (left) or inhibitors (NVP-2 or importazole; right) at 12 h post-plating. Enucleation rates were measured at 12, 14, 15 and 16 h post-plating. DMSO (black; negative control) is included. Enucleation rates at hour 16 were statistically compared ns, not significant. (G) Enucleation rates of mouse orthochromatic erythroblasts following inhibitor washout into fresh medium containing DMSO (left) or inhibitors (KN-62 or importazole; right) at 12 h post-plating. Enucleation rates were measured at 12, 14, 15 and 16 h post-plating. DMSO (black; negative control) is included. Enucleation rates at hour 16 were statistically compared. For E–G, *n*=5 replicates from two independent experiments. ns, not significant; *P≤0.05; **P<0.01; ***P≤0.001; ****P≤0.0001 (two-way ANOVA with Tukey's multiple comparisons test). In E–G, colours change to match either DMSO (grey) or inhibitors (coloured) after washouts. All error bars show mean±s.e.m. Scale bars: 5 μm.

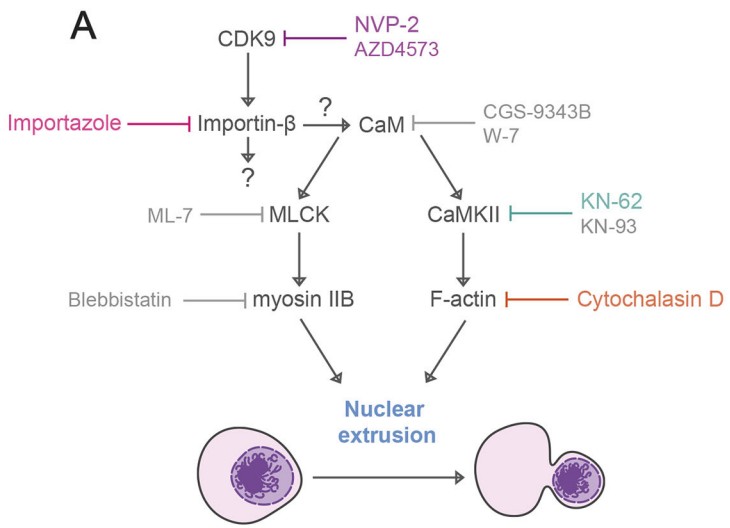

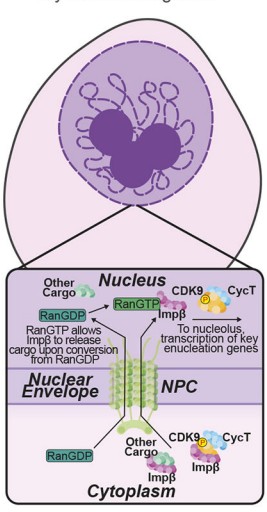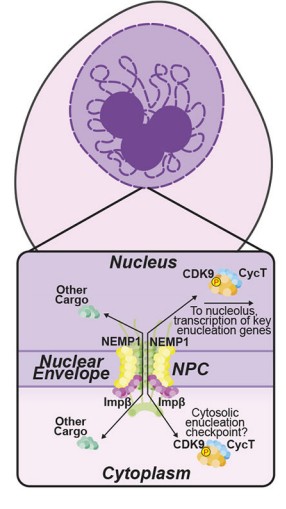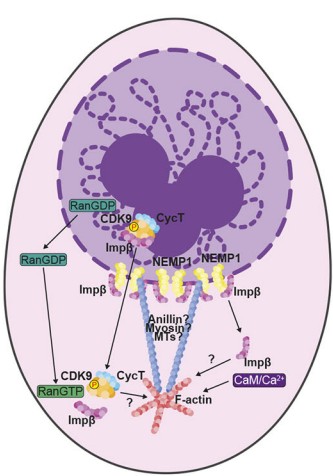

**Fig. 6. Proposed non-mutually exclusive models for the role of CDK9 and importin-β in regulating erythroid enucleation prior to CaM and Ca²⁺ signalling.** (A) Order of action model of CDK9 activity to achieve nuclear extrusion by downstream activation of the CaM pathway through direct or indirect activity of importin-β. CaM activation results in CaMKII and MLCK activation which in turn results in F-actin polymerisation and myosin IIB contraction, respectively, to achieve nuclear extrusion. Inhibitors shown in colours correspond to drugs used throughout this study. Inhibitors shown in grey have been previously described in this context (Wölwer et al., 2016). (B) Non-mutually exclusive models depicting potential roles for CDK9 and importin-β in enucleation. Model 1: importin-β regulates nuclear import of essential regulators of enucleation. CDK9 facilitates transcription of key enucleation genes. Model 2: importin-β acts alongside NEMP1 to facilitate essential nuclear envelope openings, allowing nucleocytoplasmic transport of CDK9 and other key enucleation regulators. Model 3: CDK9 regulates an enucleation checkpoint in the cytoplasm. importin-β, NEMP1 and Ran facilitate key signalling mechanisms between the nucleus, nuclear envelope and cytoskeleton.

that both importin-β and CDK9 are required near the membrane opposite the polarised nucleus prior to nuclear extrusion. A proposed order of the molecular events regulating enucleation is summarised in Fig. 6, alongside non-mutually exclusive potential models for the role of CDK9 and importin-β in erythroid enucleation.

## DISCUSSION
### CDK9 and importin-β facilitate cellular reorganisation prior to CaM and Ca²⁺ signalling and F-activity to achieve enucleation
Our previous work had identified CDK9 as a regulator of enucleation (Wölwer et al., 2015), and this study aimed to delineate the

mechanisms involved. In doing so, we have identified importin-β as a novel regulator of enucleation and direct interactor of CDK9. How might importin-β and CDK9 coordinate erythroid enucleation? Here, we propose three non-mutually exclusive models (Fig. 6B). The typical biological function of CDK9 is well characterised, where it acts as the kinase component and small subunit of P-TEFb to regulate RNA Pol II transcriptional elongation by phosphorylating the C-terminal domain (CTD) of RNA Pol II and negative regulators DRB sensitivity-inducing factor (DSIF) and negative elongation factor (NELF) (Anshabo et al., 2021; Gilchrist et al., 2008; Paparidis et al., 2017; Wada et al., 1998). Our previous study had shown that blocking the activity of RNA Pol II in late erythropoiesis does not

arrest enuclation (Wölwer et al., 2015), suggesting a new role for CDK9 in enucleation independent of RNA Pol II. Neither DSIF or NELF have been shown to interact with or modulate the activity of anything other than RNA Pol II, and we did not identify them in our proteomics search, therefore we did not investigate their potential role in enucleation but at this stage cannot rule out a possible involvement (Aoi et al., 2020; Decker, 2021; Deng et al., 2022). Considering the localisation of CDK9 during enucleation and the phenotype of CDK9 inhibitor-induced arrest, we conclude that CDK9 must be acting within the cytoplasm prior to nuclear extrusion; however, it remains plausible that CDK9-mediated transcription via RNA Pol II is essential for enucleation and that importin-β might play a role in the nucleocytoplasmic location of CDK9 (Fig. 6B, Model 1).

NEMP1 was identified across all replicates in our CDK9 co-IP-MS search, and plays an essential role in enucleation by modulating nuclear envelope openings that are essential for enucleation; NEMP1-knockout mice also show anaemia due to apoptosis of polychromatic erythroblasts (Hodzic et al., 2022). Considering both NEMP1 and importin-β are key components of the Ran nuclear import pathway (Shibano et al., 2015), it is plausible that the function of importin-β in enucleation is related to that of NEMP1. Interestingly, NEMP1 has been shown to support the mechanical stiffness of the nuclear envelope in some settings (Tsatskis et al., 2020), potentially implicating NEMP1 (and by association importin-β) in the mechanical processes involved in extrusion of the nucleus during enucleation. Considering CDK9 and importin-β were found to physically interact with NEMP1 in the context of erythropoiesis, and that importin-β inhibition mimics the observed phenotype in NEMP1-knockout mice (Hodzic et al., 2022), we postulate that importin-β, NEMP1 and possibly other members of the Ran nucleocytoplasmic import complex regulate changes to the nuclear envelope to assist in the shuttling of key enucleation regulators, including CDK9, between the cytoplasm and nucleus prior to nuclear extrusion (Fig. 6B, Model 2).

Finally, our previous work has identified a role for $Ca^{2+}$ signalling in enucleation via the CaM pathway, and the role of F-actin is well documented (Newton et al., 2024; Wölwer et al., 2016). More recently, Zhang et al. (2025) have described and support the notion of finely tuned CaM and $Ca^{2+}$ signalling during erythropoiesis and enucleation (Zhang et al., 2025). Here, we examined the order in which CDK9 and importin-β might act to regulate enucleation with respect to $Ca^{2+}$ signalling and F-actin activity and found both to function upstream. The CaM pathway is implicated in both myosin and F-actin activation through the activity of myosin light chain kinase (MLCK), and $Ca^{2+}$ and CaMKII, respectively (Mizuno et al., 2008; Okamoto et al., 2007). CaM signalling has been shown to regulate CDK9 T-loop phosphorylation, which is required for its activation, which might suggest a coordinated role for CaM signalling in separate steps in enucleation involving CDK9 and the cytoskeleton (Ramakrishnan and Rice, 2012). Additionally, $Ca^{2+}$ signalling, and intracellular $Ca^{2+}$ levels have been shown to block nuclear import by altering the activity and localisation of RanGTP and importin-β (Kaur et al., 2014; Sweitzer and Hanover, 1996). Strikingly, blocking enucleation with an actin inhibitor or importin-β inhibitor both caused a reproducible and consistent accumulation of phosphorylated CDK9 near the junction of enucleation. This may be due to the aggregation of activated CDK9 at a critical site of activity, which we speculate may be the proposed 'enucleosome' (Nowak et al., 2017), therefore priming the cells for nuclear extrusion following release of F-actin or importin-β inhibition. We propose that CDK9 and importin-β might function as part of a checkpoint in the final stages of enucleation, in conjunction with an influx of $Ca^{2+}$ and subsequent CaM signalling to achieve nuclear extrusion (Fig. 6B, Model 3).

## Characterisation of the CDK9 interactome in terminally differentiating erythroblasts

This study provides new insights into the CDK9 interactome during erythroblast development. Importantly, we identified several components of the Ran nuclear import pathway, including importin-β and RanGTP itself, which are associated directly with CDK9 in both undifferentiated and differentiated HUDEP-2 cells. How might RanGTP act to regulate enucleation? RanGTP acts as a master controller for nucleocytoplasmic transport through the nuclear pore complex (NPC) by interactions with importins and exportins (Cavazza and Vernos, 2016). CDK9 is known to shuttle between the nucleus and cytoplasm, and other interactors of CDK9 including cyclin T and HEXIM family proteins have been shown to interact with importin-α to facilitate transport through the NPC (Czudnochowski et al., 2010; Napolitano et al., 2002). CDK9 has previously been shown to interact with nuclear pore components RANGAP1, RANBP2 and RGPD3 (Ramakrishnan et al., 2012), which we also identified as part of the RanGTP complex that interacts with CDK9 in erythroid cells. However, the previous study only focused on localisation of CDK9–cyclin T1 to the nuclear pore and did not examine the role of Ran and importin-β in CDK9–cyclin T1 nuclear import. Of note, exportin 7 (Xpo7), which can facilitate nuclear export through the RanGTP pathway, is essential for histone export into the cytoplasm during the nuclear condensation phase of enucleation (Hattangadi et al., 2014; Mingot et al., 2004). It will be interesting to examine the relationship between the RanGTP pathway and Xpo7 in the control of histone export and how CDK9 might be involved in this process. Although this study offers novel insights into the potential regulation of CDK9 and cyclin T1 nucleocytoplasmic transport via importin-β interactions, we have clearly shown that importin-β interacts with CDK9 in erythroblasts and is required for enucleation upstream of CaM and $Ca^{2+}$ signalling and F-actin activity.

What else might the CDK9 and importin-β interactome tell us about enucleation? Besides nuclear import, importin-β has been shown to modulate the permeability and structure of the NPC through direct interaction with NPC protein Nup153 (Lowe et al., 2015), highlighting a potential role in modifying nuclear structure to facilitate enucleation. Additionally, importin-β is implicated in wider transport roles, where it can regulate kinesin-4 motor activity in plant cells (Ganguly et al., 2018) and can directly bind to certain cargo, such as the parathyroid hormone (PTH)-related protein (PTHrP), to facilitate microtubule-dependent transport (Lam et al., 2002). Importantly, importin-β is implicated in the regulation of anillin during cytokinesis (Beaudet et al., 2020); anillin acts to scaffold components of the actomyosin ring, including F-actin, myosin and RhoA (Piekny and Glotzer, 2008), all of which are essential for nuclear extrusion during enucleation. Examining whether anillin is required for enucleation and how this may relate to importin-β function will be key to investigating this link.

In addition to the Ran import pathway complex, we also identified several other new interactors of CDK9. Golgin A2 (GOLGA2), which was observed across all sample sets and was enriched in CDK9 pulldowns, acts to maintain Golgi structure through stacking of Golgi cisternae and has been investigated as a therapeutic target in cancer due to its relationship with autophagy (Chang et al., 2012; Nakamura et al., 1995). As autophagy is essential for erythroid terminal differentiation, it will be important to examine the relationship between CDK9 and golgin A2-mediated autophagy mechanisms during enucleation. Vimentin (VIM), which is an intermediate filament protein implicated in the function of many cells types (Danielsson et al., 2018) was also identified across all sample sets,

Journal of Cell Science

and has been previously identified in a similar CDK9 co-IP-MS interactome set (Yang et al., 2015). Yang et al. (2015) defined a CDK9 interactome in human A549 pulmonary epithelial cells, and some of these proteins are also identified in our data (Yang et al., 2015). However, most proteins in our identified interactome are new, likely reflecting the specialised role CDK9 might play in erythroid development.

Of note, we identified a large complex involved in MCM and the DNA damage response. Although CDK9 has been shown to play a role in maintaining genome integrity in partnership with cyclin K (Yu and Cortez, 2011), we did not detect cyclin K in our proteomics searches, and it is therefore possible that this mechanism extends to erythroblasts through a different cyclin. This should be investigated further. Furthermore, a role for CDK9 in mRNA splicing has been described (Hu et al., 2021), and with the identification of mRNA splicing mechanisms, polyadenylation factors and several members of the messenger ribonucleoprotein complex assembly, also known as mRNP granules (Buchan, 2014), in human erythroblasts, new insights could be gained on the role of CDK9 in mRNA processing in erythropoiesis. Importantly, we also identified a large complex involved in actin filament capping and binding, including CAPZA1, CAPZA2 and CAPZB, among others, which might point to how CDK9 might interact with actin during enucleation processes. Additionally, we identified tropomodulin-1, which has been observed to colocalise with F-actin and non-musical myosin IIB during enucleation (Nowak et al., 2017), and Cdc42, which regulates essential polarisation of the nucleus prior to nuclear extrusion (Ubukawa et al., 2020). The relationship between CDK9 and the regulation of actinomyosin polymerisation should be investigated further, using genetic models for proximity labelling and functional assays to determine how CDK9 activity may be involved in modulating actinomyosin activity in the context of erythroid terminal differentiation. Although we do not fully understand how CDK9, importin-β and downstream effectors of enucleation directly interact to coordinate enucleation, our studies provide insight into a series of events that must occur before F-actin mediated nuclear extrusion and presents possible new interactors of CDK9 and importin-β in the context of erythropoiesis and erythroid enucleation.

## MATERIALS AND METHODS

### Materials
Antibodies, inhibitors and other reagents are listed in Table S1. pBABE-Flag-Cdk9-IRES-eGFP, pBABE-Flag-Cdk9-T186A-IRES-eGFP and pBABE-Flag-Cdk9-D167N-IRES-eGFP were from Addgene (deposited by Andrew Rice; Addgene plasmid #28096, RRID: Addgene_28096; Addgene plasmid #28097, RRID: Addgene_28097; Addgene plasmid #28098, RRID: Addgene_28098); pBABE GFP was from Addgene (deposited by William Hahn; Addgene plasmid #10668, RRID: Addgene_10668).

### Isolation of orthochromatic erythroblasts
All animal procedures were approved by the La Trobe University Animal Ethics Committee. Splenic orthochromatic erythroblasts were isolated from C57Bl/6 mice (female and male) between 6 and 14 weeks of age by phenhylhydrazine treatment and flow cytometry as previously described (Wölwer et al., 2015). Briefly, we sorted for viable (PI negative) Ter119$^+$CD44$^{low}$ cells, which have been previously validated to be primarily orthochromatic erythroblasts (Wölwer et al., 2015). In normal conditions, we regularly observed ~60% enucleation 5 h after flow cytometric sorting, which remained stable for up to 16 h.

### HUDEP-2 and BEL-A cell culture
HUDEP-2 and BEL-A cells were cultured as previously described (Trakarnsanga et al., 2017); cells were obtained from Dr Ryo Kurita (RIKEN BioResource Research Centre, Japan) and Prof. Jan Frayne

(University of Bristol, UK), respectively. HUDEP-2 and BEL-A cells were authenticated and tested for contamination prior to use.

### Flow cytometry
Enucleation of mouse orthochromatic erythroblasts was quantified using a FACSymphony A3 (Becton Dickson; Franklin Lakes, NJ, USA) running FACS Diva software (Becton Dickson) and analysed using FlowJo v10.8.1 (Becton Dickson). Mouse orthochromatic erythroblasts were stained with Hoechst 33342 and Ter119-Alexa Fluor 647 and assessed as previously described (Wölwer et al., 2015, 2016). Briefly, the percentage of enucleated cells was obtained by dividing the number of enucleated cells (Ter119$^+$ Hoechst$^-$) by the sum of enucleated cells and erythroblasts (Ter119$^+$ Hoechst$^+$) and multiplying by 100. Propidium iodide was used to eliminate dead cells from the analysis.

### Cytospins
$6\times10^4$ cells were centrifuged onto slides at 320 rpm for 4 min using a Cytospin 4 Centrifuge (Thermo Fisher Scientific; Waltham, MA, USA). Slides were air dried prior to fixation in methanol and subsequently stained using RapidDiff as per the manufacturer's guidelines. HUDEP-2 and BEL-A enucleation rates, and cell morphologies were quantified manually by images captured using an Olympus IX81 microscope using 100×/1.4NA oil objective running cellSens Dimension software (Evident/Olympus Life Sciences; Tokyo, Japan).

### Immunofluorescence confocal microscopy
Cells were prepared for immunofluorescence microscopy as previously described (Smith et al., 2018) with minor alterations. Briefly, cells were collected by centrifugation following overnight incubation with or without inhibitors, washed in PBS and immediately fixed in 4% paraformaldehyde in PBS overnight at 4°C. Cells were subsequently washed three times in PBS and permeabilised in PBS containing 0.3% Triton X-100 for 15 min at room temperature (RT), before blocking with PBS containing 3% BSA and 1% donkey serum for 2 h at RT. Cells were incubated with primary antibodies in blocking buffer overnight at 4°C, washed three times in PBS and incubated with secondary antibodies and 1 µg/ml DAPI in blocking buffer for 2 h. Cells were washed three times in PBS before centrifugation onto poly-D-lysine-treated microscopy slides using a cytospin centrifuge and subsequently mounted in ProLong Gold antifade reagent. Slides were imaged using a Zeiss LSM800 confocal microscope using 63×/1.4NA oil objective running Zeiss Zen Blue software (Zeiss; Oberkochen, Germany). Nuclear to cytoplasmic signal ratios were calculated using Fiji (ImageJ; Schindelin et al., 2012). Our custom nuclear-to-cytoplasmic ratio ImageJ macro is available on GitHub (https://github.com/La-Trobe-Bioimaging-Platform/La-Trobe-Bioimaging-Platform-Scripts/blob/main/2%20channel%20Fluorescence%20Ratio%20Calculator).

### Retroviral transduction of HUDEP-2 cells
Retrovirus was generated by calcium phosphate transfection of HEK293T cells as previously described (Jordan et al., 1996). Briefly, at 12 h post-transfection, transfection medium was replaced with HUDEP-2 expansion medium. Viral medium was collected after 24 h, and HUDEP-2 cells were seeded at $3\times10^5$ cells/ml in viral growth medium for 24 h. Transduced cells were isolated using FACS Aria Fusion (Becton Dickson; Franklin Lakes, NJ) running FACS Diva software (Becton Dickson; Franklin Lakes, NJ) by sorting for GFP$^+$ cells.

### CDK9 co-immunoprecipitation
Co-immunoprecipitation studies were performed using the Pierce Classic Magnetic IP/Co-IP Kit (Thermo Fisher Scientific) as per the manufacturer's instructions. Modifications were as follows; cells were lysed in undifferentiated state or on day 6 of differentiation in NETN buffer (100 mM NaCl, 20 mM Tris-HCl pH 8.0, 0.5 mM EDTA and 0.5% v/v NP-40) supplemented with Complete Mini protease inhibitor and PhosSTOP protease inhibitor as per the manufacturer's guidelines (Roche, Basel, Switzerland). Protein extracts were quantified using the DC Protein Assay (Bio-Rad, Hercules, CA, USA). Three separate replicate immune complexes were prepared by combining 10 µg of CDK9 (F-6) or 10 µg mouse IgG

Journal of Cell Science

isotype control antibody with 1 mg of cell lysate and incubated while mixing at 4°C overnight. Immunoprecipitation was performed overnight while mixing at 4°C using 0.3 mg of freshly washed A/G magnetic beads. On-bead trypsin digestion was carried out as previously described (Antonicka et al., 2020). Briefly, beads were washed three times and resuspended in 20 mM ammonium bicarbonate (pH 8) before addition of Tris (2-carboxyethyl) phosphine to a final concentration of 5 mM and incubated at 45°C for 30 min to achieve reduction of cysteine disulphide bonds. Iodoacetamide was added to a final concentration of 20 mM and incubated in the dark for 10 min to achieve alkylation. Beads were pelleted and resuspended in 20 mM ammonium bicarbonate (pH 8) with 2.5 µg/ml of trypsin before incubation at 37°C overnight. Beads were pelleted and the supernatant collected, before rinsing the beads in 20 mM ammonium bicarbonate (pH 8) and recombining with the supernatant. Samples were dried in a centrifugal evaporator and desalted before liquid chromatography tandem mass spectrometry (LC-MS/MS).

### Liquid chromatography and mass spectrometry analysis

The tryptic peptides were separated on a Thermo Ultimate 3000 RSLC nano UHPLC system and analysed on a Thermo Q-Exactive HF Orbitrap mass-spectrometer (Thermo Fisher Scientific). Peptides were loaded onto a PepMap C18 5 µm 2 cm trapping column (Thermo Fisher Scientific) and washed at 5 µl/min for 6 min using Buffer C [0.1% (v/v) trifluoroacetic acid and 2% (v/v) acetonitrile (ACN)] before switching the pre-column in line with the analytical column held at 55°C (nanoEase $m/z$ Peptide BEH C18 Column, 1.7 µm, 130 Å and 75 µm ID×25 cm; Waters Corporation, Milford, MA, USA). The separation of peptides was performed at 250 nl/min using a linear gradient of buffer A [0.1% (v/v) formic acid, 2% (v/v) ACN] and buffer B [0.1% (v/v) formic acid, 80% (v/v) ACN], starting at 12% buffer B to 30% over 54 min, then rising to 50% B over 10 min followed by 95% B in 6 min. The column was then cleaned for 4 min at 95% B and then the percentage of B was brought down to 2% over 5 min. The column was equilibrated with 2% B for 15 min. Blanks were run between sample injections. MS data were collected in the data-dependent acquisition (DDA) mode over a period of 90 min. MS1 scan parameters were: 60,000 resolution, $m/z$ range of 350–1500, AGC target $3\times10^6$, maximum ion injection time 30 ms. The top seven ions were fragmented per cycle using a normalised collision energy of 28 and dynamic exclusion was carried out for 25 s. The isolation window of the quadrupole for precursor isolation was 1.4 $m/z$. MS2 scan parameters were: 60,000 resolution, AGC target $1\times10^5$, maximum ion injection time 110 ms. Lock mass was set to 445.1200 for internal mass calibration.

### Protein database search and analysis

The protein database searches were conducted using Sequest HT search engine via the Proteome Discoverer 2.4 software suite (Thermo Fisher Scientific). Spectra were matched against the *Homo sapiens* reference proteome downloaded from Uniprot (The UniProt Consortium, 2023). Precursor tolerance was set to 20 ppm and fragment tolerance to 0.05 Da. Two missed trypsin cleavages were permitted. The included static modification was carbamidomethyl of cysteine, and dynamic modifications were oxidation of methionine, acetylation of the protein N-terminus, and deamidation of asparagine and glutamine. Validation false discovery rate (FDR) thresholds were 0.01 for strict and 0.05 for relaxed criteria at PSM, peptide and protein levels. Label-free quantification was performed on precursor ion area. Proteins that were present in all three replicates with a minimum three-fold enrichment compared to the IgG controls were considered relevant. Results, including generation of Venn diagrams and heatmaps, were analysed using FunRich v3.1.4 (Fonseka et al., 2021). Interaction and gene ontology (GO) analysis was performed using the STRING database (Szklarczyk et al., 2023). The mass spectrometry proteomics data have been deposited to the ProteomeXchange Consortium via the PRIDE (Perez-Riverol et al., 2025) partner repository with the dataset identifier PXD072077.

### Western blotting

Cell lysates were isolated using NETN lysis buffer (250 mM NaCl, 5 mM EDTA, 50 mM Tris-HCl pH 8.0, 0.5% NP-40 in distilled water) containing Complete mini protease inhibitor (Roche) and PhosSTOP phosphatase inhibitor (Roche; Basel, Switzerland). Protein levels were quantified using a DC protein assay (Bio-Rad), denatured using SDS loading buffer and

subsequently resolved on standard SDS-PAGE gels before western transfer onto polyvinylidene difluoride membrane (Merck Millipore, Burlington, MA, USA). Membranes were blocked in 3% BSA in 0.2% Tween-20 in PBS for 1 h, then incubated overnight with primary and secondary antibodies, respectively. Membranes were visualised using an Odyssey infrared imager (LiCor Biosciences; Lincoln, NE). Antibodies and concentrations used are listed in Table S1.

### Statistical analysis

Statistical calculations and analysis were performed using GraphPad Prism version 9.2.0 (GraphPad Software, San Diego, CA). Statistical differences were analysed using one- or two-way ANOVA tests, with Dunnett's or Tukey's tests for multiple comparisons. Detailed statistics are described in the figure legends. $n$=number of cells counted, or number of replicates. $P$-values below 0.05 were considered statistically significant.

### Acknowledgements

We acknowledge Dr Margaret Veale and the La Trobe Bioimaging Platform for flow cytometry and optical microscopy. We acknowledge Dr Pierre Faou, Dr Rohan Lowe, Dr Keshava Datta and the La Trobe Proteomics and Metabolomics Platform for mass spectrometry and proteomics. We acknowledge the La Trobe Animal Research & Training Facility (LARTF) for supporting animal work. We thank Dr Ryo Kurita from the Riken BioResource Research Center (Tsukuba, Japan) for providing HUDEP-2 cell lines, and Prof. Jan Frayne from the University of Bristol (Bristol, UK) for providing BEL-A cell lines.

### Competing interests

The authors declare no competing or financial interests.

### Diversity and inclusion

This work was completed on Wurundjeri land of the Kulin nation, and we pay our respects to the Elders past and present.

### Author contributions

Conceptualization: S.M.R., P.O.H.; Methodology: L.M.N., K.Y.B.L., D.Y.A., C.B.W., P.O.H.; Software: C.J.J.; Validation: L.M.N., K.Y.B.L., D.Y.A.; Formal analysis: L.M.N.; Investigation: L.M.N., K.Y.B.L., D.Y.A.; Resources: S.M.R., E.D.H., P.O.H.; Data curation: L.M.N., C.J.J., P.O.H.; Writing – original draft preparation: L.M.N., P.O.H.; Writing – review and editing: L.M.N., K.Y.B.L., D.Y.A., C.B.W., S.M.R., P.O.H.; Visualization: L.M.N.; Supervision: S.M.R., E.D.H., P.O.H.; Project administration: L.M.N., P.O.H.; Funding acquisition: E.D.H., P.O.H.

### Funding

This work was supported by National Health and Medical Research Council (NHMRC) Project grant APP1103858 (P.O.H.) and Australian Research Council (ARC) Discovery project grant DP190103634 (P.O.H. and E.D.H.). L.M.N. was supported by an Australian Government Research Training Program (RTP) scholarship. P.O.H. was supported by an NHMRC Fellowship (APP1079133). Open Access funding provided by La Trobe University. Deposited in PMC for immediate release.

### Data and resource availability

The mass spectrometry proteomics data have been deposited to the ProteomeXchange Consortium via the PRIDE (Perez-Riverol et al., 2025) partner repository with the dataset identifier PXD072077. All other relevant data and details of resources can be found within the article and its supplementary information.

### Peer review history

The peer review history is available online at https://journals.biologists.com/jcs/lookup/doi/10.1242/jcs.264385.reviewer-comments.pdf

### Special Issue

This article is part of the Special Issue 'Cell Biology of the Nucleus', guest edited by Abby Buchwalter. See related articles at https://journals.biologists.com/jcs/issue/139/12.

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
