## [Peer Review File · Journal of Cell Science]

CDK9 interacts with a RanGTP-importin- β complex to regulate erythroid enucleation

Lucas M. Newton, Krystle Y. B. Lim, Donia Y. Abeid, Christina B. Wölwer, Chad J. Johnson, Sarah M. Russell, Edwin D. Hawkins and Patrick O. Humbert
DOI: 10.1242/jcs.264385

Editor: Megan King

Review timeline

Original submission:	14 August 2025
Editorial decision:	18 September 2025
First revision received:	18 December 2025
Accepted:	12 February 2026

Original submission

First decision letter

MS ID#: jcs.264385

MS TITLE: CDK9 interacts with a RanGTP-NEMP1-Importin- β complex to regulate erythroid enucleation

AUTHORS: Lucas Newton; Krystle Lim; Christina Wölwer; Donia Abeid; Chad Johnson; Sarah Russell; Edwin Hawkins; Patrick Humbert

ARTICLE TYPE: Transfer

Dear Dr Newton,

We have now reached a decision on the above manuscript.

To see the reviewers' reports and a copy of this decision letter, please go to:

As you will see, the reviewers raise a number of substantial criticisms that prevent me from accepting the paper at this stage. They suggest, however, that a revised version might prove acceptable, if you can address their concerns. If you think that you can deal satisfactorily with the criticisms on revision, I would be pleased to see a revised manuscript. We would then return it to the reviewers.

Reviewer 1

Advance summary and potential significance to field

The manuscript entitled "CDK9 interacts with a RanGTP-NEMP1-Importin- β complex to regulate erythroid enucleation" presents an interesting study investigating the interaction between CDK9 and the RanGTP-NEMP1-Importin- β complex during erythroid terminal differentiation. The work provides novel mechanistic insights into CDK9 as a regulator of erythroid enucleation, independent of its canonical role in RNA polymerase II transcription. The findings have potential translational significance, given the clinical development of CDK9 inhibitors and the ongoing efforts to generate

red blood cells *ex vivo*. While the study addresses an important biological process with translational relevance, the evidence presented falls short of substantiating the central claims.

Comments for the author

The manuscript frames Nemp1 as a mechanistic driver of enucleation with CDK9 and RNAGTP. However, the role of NEMP1 remains primarily inferred from previous reports rather than demonstrated directly in this study. Its inclusion in the pathway is based solely on proteomics detection and prior reports. Proteomic findings are not validated with co-IP/Western blot. Figures do not fully reflect NEMP1 localization or phenotypes, which diminishes the clarity of the proposed models. The authors did not clarify whether NEMP1 was consistently detected across replicates in proteomics. The biological vs technical repeats are not clearly stated in the co-IP/Mass spec experiment.

The mechanistic order of events between CDK9, Importin- β , NEMP1, and CaM/F-actin remains unresolved. Inhibitor washout experiments convincingly place CDK9 and Importin- β upstream of F-actin, but the order relative to CaM/Ca²⁺ signaling remains ambiguous. The inhibitor washout assays are useful, but cannot establish a sequential hierarchy, since NEMP1 was not directly tested.

Inhibitor-induced arrests (CDK9, Importin- β , F-actin) show similar phenotypes. The arrest phenotypes caused by CDK9, Importin- β , and actin inhibition appear highly similar, yet the authors interpret them as evidence of a distinct order of function. More refined phenotypic characterization (e.g., nuclear envelope morphology, actomyosin structure, nuclear stiffness assays) would strengthen the study.

The current discussion briefly mentions previous published studies indicating NEMP1's roles in 1) stiffness, and 2) in Ran recruitment but the integration is superficial and explaining how the information models would strengthen the paper. The authors present three different models for how CDK9 and importin- β may regulate enucleation prior to CaM/Ca²⁺ signaling, but none of these models are experimentally tested. Without such evidence, the title and overall content of the manuscript should be revised to reflect descriptive observations rather than untested mechanistic models.

Detailed points.

1. CDK9 inhibitor treatments (Figure 2B and 2C):

The authors use the CDK9 inhibitors NVP-2 and AZD4573 in orthochromatic mouse erythroblasts. It is essential to assess cell viability under these conditions to exclude potential cytotoxic effects that could confound the enucleation results.

2. Cytochalasin D concentration (Figure 2):

The manuscript compares CDK9 inhibition with cytochalasin D (CytoD)-induced late-stage enucleation blockade. The data show that 300 nM CytoD significantly impairs enucleation (Figure 2D). However, the authors used 1 μ M CytoD in the quantification of morphological results. The rationale for selecting such a high concentration should be clarified, as higher doses may introduce cytotoxicity and obscure interpretation.

3. Vehicle controls (Figure 3A, 3C):

DMSO (vehicle) controls should be included in Figure 3A and 3C to strengthen the interpretation of inhibitor-specific effects.

4. Validation of CDK9 interactors:

The proteomics screen identifying novel CDK9 interactors in the context of erythroid differentiation is a strength of the study. However, the manuscript lacks independent validation of these interactions. Co-immunoprecipitation followed by Western blot analysis should be performed to confirm the proteomics findings.

5. Importazole concentration (Figure 5A):

The authors used 10 μ M importazole to assess its effect on enucleation in mouse orthochromatic erythroblasts. The rationale for choosing this concentration over lower doses should be provided, especially in light of potential off-target or cytotoxic effects.

6. Vehicle controls (Figure 5D):

DMSO (vehicle) controls should be included in Figure 5D to ensure the observed effects are specific to importazole treatment.

Minor points:

1. Use "NEMP1" consistently rather than alternating with "TMEM194A."
2. Correct typographical issues in the discussion (e.g., repeated "importin- β importin- β ").

Reviewer 2

Advance summary and potential significance to field

Major comments:

The role of CDK9 in erythroid enucleation has been reported before, therefore the novelty of the current manuscript is not high. However, CDK9's interaction with NEMP1 is novel. The authors need more evidence to address the functional impact of this interaction.

1. The entire manuscript lacks genetic evidence. The authors should knockout CDK9 using CRISPR (NOT shRNA) to confirm their findings.
2. The correct flow cytometry assays should be used to quantify enucleation. There are no quantification of enucleated reticulocytes in the current manuscript.
3. What is the functional consequence of CDK9 and NEMP1 interaction? Across the manuscript, there are no studies investigating the nuclear opening formation. This should be done.

First revision

Author response to reviewers' comments

Reviewer 1: SUMMARY OF THE ADVANCE MADE IN THIS PAPER AND ITS POTENTIAL SIGNIFICANCE TO THE FIELD

The manuscript entitled "CDK9 interacts with a RanGTP-NEMP1-Importin- β complex to regulate erythroid enucleation" presents an interesting study investigating the interaction between CDK9 and the RanGTP-NEMP1-Importin- β complex during erythroid terminal differentiation. The work provides novel mechanistic insights into CDK9 as a regulator of erythroid enucleation, independent of its canonical role in RNA polymerase II transcription. The findings have potential translational significance, given the clinical development of CDK9 inhibitors and the ongoing efforts to generate red blood cells *ex vivo*. While the study addresses an important biological process with translational relevance, the evidence presented falls short of substantiating the central claims.

SUGGESTIONS TO AUTHORS

The manuscript frames Nemp1 as a mechanistic driver of enucleation with CDK9 and RNAGTP. However, the role of NEMP1 remains primarily inferred from previous reports rather than demonstrated directly in this study. Its inclusion in the pathway is based solely on proteomics detection and prior reports. Proteomic findings are not validated with co-IP/Western blot. Figures do not fully reflect NEMP1 localization or phenotypes, which diminishes the clarity of the proposed

models. They authors did not clarify whether NEMP1 was consistently detected across replicates in proteomics. The biological vs technical repeats are not clearly stated in the co-IP/Mass spec experiment.

The mechanistic order of events between CDK9, Importin- β , NEMP1, and CaM/F-actin remains unresolved. Inhibitor washout experiments convincingly place CDK9 and Importin- β upstream of F-actin, but the order relative to CaM/Ca²⁺ signaling remains ambiguous. The inhibitor washout assays are useful, but cannot establish a sequential hierarchy, since NEMP1 was not directly tested.

Inhibitor-induced arrests (CDK9, Importin- β , F-actin) show similar phenotypes. The arrest phenotypes caused by CDK9, Importin- β , and actin inhibition appear highly similar, yet the authors interpret them as evidence of a distinct order of function. More refined phenotypic characterization (e.g., nuclear envelope morphology, actomyosin structure, nuclear stiffness assays) would strengthen the study.

The current discussion briefly mentions previous published studies indicating NEMP1's roles in 1) stiffness, and 2) in Ran recruitment but the integration is superficial and explaining how the information models would strengthen the paper. The authors present three different models for how CDK9 and importin- β may regulate enucleation prior to CaM/Ca²⁺ signaling, but none of these models are experimentally tested. Without such evidence, the title and overall content of the manuscript should be revised to reflect descriptive observations rather than untested mechanistic models.

Response to Reviewer 1:

We thank Reviewer 1 for the insightful comments and suggestions regarding our manuscript.

We agree with both reviewers (see our original key points section, now labelled as summary statement) that the key discovery in our study is the identification of Importin- β as a regulator of enucleation. Our findings, combined with the literature, indicate a role for NEMP1, but we agree with both reviewers that this is not the key finding, and it was not our intention for this finding to be at the forefront of our study. To more effectively direct the focus to the interaction between CDK9 and the RanGTP-Importin- β complex, we have removed mention of NEMP1 from the manuscript title and abstract and focused more closely on observations related to Importin- β in the introduction and results section. We have highlighted the broader implications of likely interactions between CDK9, RanGTP, Importin- β and NEMP1 in the discussion only.

Regarding the mechanistic order of events, we are confident that our drug washout assays accurately reflect the sequential nature in which Importin- β and CDK9 act upstream of F-actin during enucleation. However, we do agree that since NEMP1 was not directly tested, we cannot infer where in this order of events NEMP1 may act. We have addressed this by focusing on the inhibitor washouts shown in this study and have provided some insights and speculation in the discussion section only. We have also altered the wording throughout the results section to make clear that any statement related to the order of events is based on the results of inhibitor washout assays rather than phenotyping of erythroblasts under enucleation arrest.

We thank the reviewer for suggestions related to gaining more mechanistic insights through more detailed phenotypic analysis (such as nuclear stiffness, actomyosin structure etc). We agree that these types of experiments would greatly enhance our understanding of the underlying mechanism and the role of CDK9 and Importin- β , however these types of experiments are unfortunately beyond the scope of our study. We sincerely hope that future research can build on our findings and conduct such interesting mechanistic experiments. Accordingly, we have altered the wording throughout parts of the manuscript to better reflect the functional/descriptive characterisation of our findings and downplay inferred mechanistic models.

Detailed points.

1. CDK9 inhibitor treatments (Figure 2B and 2C):

The authors use the CDK9 inhibitors NVP-2 and AZD4573 in orthochromatic mouse erythroblasts. It is essential to assess cell viability under these conditions to exclude potential cytotoxic effects that could confound the enucleation results.

To ensure that the inhibitors used throughout this study did not cause overt toxicity, we always included propidium iodide (PI) staining in flow cytometry analysis to exclude non-viable cells from analysis and to monitor overall cell viability. Cell viability was minimally affected by inhibitors used in the experiments shown throughout this study, and to better reflect this, we have created a new supplementary figure (Supplementary Figure 3) showing PI staining and cell-viability quantification for all inhibitors used.

2. Cytochalasin D concentration (Figure 2):

The manuscript compares CDK9 inhibition with cytochalasin D (CytoD)-induced late-stage enucleation blockade. The data show that 300 nM CytoD significantly impairs enucleation (Figure 2D). However, the authors used 1 μ M CytoD in the quantification of morphological results. The rationale for selecting such a high concentration should be clarified, as higher doses may introduce cytotoxicity and obscure interpretation.

We agree that 300nM CytoD impairs erythroblast enucleation as well as the higher 1 μ M dosage. However, we chose to use the same 1 μ M dosage that had been historically used throughout similar studies from our laboratory (Wölwer et al, 2015, Wölwer et al, 2016) and others (Wang et al, 2012). We would also note that 1 μ M CytoD does not cause adverse toxicity in erythroblasts and its effects are highly reversible.

3. Vehicle controls (Figure 3A, 3C):

DMSO (vehicle) controls should be included in Figure 3A and 3C to strengthen the interpretation of inhibitor-specific effects.

Thank you, we agree that it was an oversight not to include DMSO controls for the imaging shown in Figure 3A. Additionally, we did not make clear that the images shown in Figure 3B are indeed DMSO vehicle control samples. We always included DMSO in all untreated samples to ensure comparability with inhibitor treatments. To address this, we have included extra images in Figure 3A which correspond to DMSO-treated controls and changed the labelling in Figure 3B to clearly identify this to the readers.

4. Validation of CDK9 interactors:

The proteomics screen identifying novel CDK9 interactors in the context of erythroid differentiation is a strength of the study. However, the manuscript lacks independent validation of these interactions. Co-immunoprecipitation followed by Western blot analysis should be performed to confirm the proteomics findings.

In response to the comments in the main response and this point regarding lack of Co-IP Western Blot to accompany the Co-IP MS data, improved workflows such as the one we adopted here have reduced artefacts and so removed the need for confirmation by western blot. This is routine for such studies, as indicated in these references (Morris et al, 2014; Ewing et al, 2007, Lukauskas et al, 2024; Jensen et al, 2014).

5. Importazole concentration (Figure 5A):

The authors used 10 μ M importazole to assess its effect on enucleation in mouse orthochromatic erythroblasts. The rationale for choosing this concentration over lower doses should be provided, especially in light of potential off-target or cytotoxic effects.

We assessed the effect of various concentrations of importazole on enucleating erythroblasts and found a dosage of 10 μ M to have the most reliable effect without significantly increased toxicity (approx. 5% increase in non-viable cells compared to the DMSO control, and <3% increase in non-viable cells compared to the lowest tested dosage 300nM - this data is now included in the new Supplementary Figure 3). Additionally, all non-viable (PI+) cells are excluded from further analysis.

To make it clearer to the readers why we have chosen a 10 μ M dosage, we have amended the manuscript to include references to other studies using importazole to inhibit Importin- β , which

report the use of higher concentrations in *in vitro* experiments, such as 40 μ M in the original publication (Soderholm et al, 2012) and other higher or similar dosages (Yoshino et al, 2021; Bird et al, 2013; Kublun et al, 2014).

6. Vehicle controls (Figure 5D):

DMSO (vehicle) controls should be included in Figure 5D to ensure the observed effects are specific to importazole treatment.

Like point 3, this was an oversight regarding the labelling of untreated cells in Figure 5C. We have altered the labelling on the figure itself to better reflect that the cells shown are indeed DMSO vehicle controls.

Minor points:

1. Use "NEMP1" consistently rather than alternating with "TMEM194A."

Thank you. We have included TMEM194A in the first mention but used only NEMP1 for all subsequent mentions.

2. Correct typographical issues in the discussion (e.g., repeated "importin- β importin- β ").

Thank you. We have fixed various typographical issues in the discussion and thank the reviewer for helping to identify them.

Reviewer 2: SUMMARY OF THE ADVANCE MADE IN THIS PAPER AND ITS POTENTIAL SIGNIFICANCE TO THE FIELD

SUGGESTIONS TO AUTHORS

Major comments:

The role of CDK9 in erythroid enucleation has been reported before, therefore the novelty of the current manuscript is not high. However, CDK9's interaction with NEMP1 is novel. The authors need more evidence to address the functional impact of this interaction.

We thank Reviewer 2 for the insightful comments. As discussed in our response to Reviewer 1, we have now made more clear the novelty of the findings.

To address this issue, we have removed NEMP1 from the manuscript title and abstract and have downplayed the association between CDK9 and NEMP1 in our introduction and results sections to focus more closely on observations related to Importin- β . However, we believe that the link between CDK9, RanGTP, Importin- β and NEMP1 is a key discussion point and is the likely next step in investigating this process, and so we have kept detailed mention of NEMP1 to the discussion only.

1. The entire manuscript lacks genetic evidence. The authors should knockout CDK9 using CRISPR (NOT shRNA) to confirm their findings.

CDK9 knockout is lethal in all cell types due to its integral role in mRNA transcription, and our attempt to knockout CDK9 in HUDEP-2 cells using CRISPR-Cas9 failed due to lethal effects on the cells. Due to this lethality, temporal control over CDK9 inhibition is critical for examining the role specifically in enucleation, which is why we chose to continue to use pharmaceutical inhibitors in our study. We are also confident that the newer, more selective and specific CDK9 inhibitors used in this study greatly reduce any chance that the observed effect is due to off-target inhibition.

2. The correct flow cytometry assays should be used to quantify enucleation. There are no quantification of enucleated reticulocytes in the current manuscript.

We are not sure which flow cytometry assays for identifying reticulocytes is being referred to here. Whilst we are familiar with flow cytometry panels that identify the stages of erythroid terminal

differentiation (stage I to V - typically using Ter119 and CD44 for mouse, combined with markers to identify earlier progenitors such as Sca-1, CD34 and c-Kit), we did not use this here as the cells in our assays were FACS-isolated orthochromatic erythroblasts. As we have published previously (Wolwer et al 2015, Wolwer et al 2016), our protocol uses Ter119 and CD44 as markers to isolate mouse splenic orthochromatic erythroblasts which undergo enucleation within 6-12 hours. Therefore, the flow cytometry assay used in this study is intended to assess enucleation only (using Ter119 and DAPI to determine the presence of absence of the nucleus) and is the most appropriate flow cytometry assay to do so.

3. What is the functional consequence of CDK9 and NEMP1 interaction? Across the manuscript, there are no studies investigating the nuclear opening formation. This should be done.

Thank you for this suggestion. As we outlined above, NEMP1 was not intended to be the focus of this study. Whilst we agree it would be insightful to examine the functional consequences of disrupting a CDK9 and NEMP1 interaction, we have only shown an association here within the CDK9 interactome of erythroblasts. Therefore, experiments to examine nuclear opening formation in the context of CDK9/Importin- β inhibition and/or NEMP-1 disruption is beyond the scope of this study.

References

- Bird, S. L., Heald, R., Weis, K. (2013). RanGTP and CLASP1 cooperate to position the mitotic spindle. *MBoC*, 24(16). <https://doi.org/10.1091/mbc.e13-03-0150>
- Ewing, R.M., Chu, P., Elisma, F. *et al.* (2007) Large-scale mapping of human protein-protein interactions by mass spectrometry. *Mol Syst Biol* 3, MSB4100134. <https://doi.org/10.1038/msb4100134>
- Jensen P, Patel B, Smith S, Sabnis R, Kaboord B. (2021). Improved Immunoprecipitation to Mass Spectrometry Method for the Enrichment of Low-Abundant Protein Targets. *Methods Mol Biol.* 2261:229-246. https://doi.org/10.1007/978-1-0716-1186-9_14
- Kublun I, Ehm P, Brehm MA, Nalaskowski MM. (2014). Efficacious inhibition of Importin α/β -mediated nuclear import of human inositol phosphate multikinase. *Biochimie.* Jul;102:117-23. <https://doi.org/10.1016/j.biochi.2014.03.001>
- Lukauskas, S., Tvardovskiy, A., Nguyen, N.V. *et al.* (2024). Decoding chromatin states by proteomic profiling of nucleosome readers. *Nature* 627, 671-679. <https://doi.org/10.1038/s41586-024-07141-5>
- Morris JH, Knudsen GM, Verschueren E, Johnson JR, Cimermancic P, Greninger AL, Pico AR. (2014). Affinity purification-mass spectrometry and network analysis to understand protein-protein interactions. *Nat Protoc.* 2014 Nov;9(11):2539-54. <https://doi.org/10.1038/nprot.2014.164>
- Soderholm, J. F., Bird, S. L., Kalab, P., Sampathkumar, Y., Hasegawa, K., Uehara-Bingen, M., Weis, K., & Heald, R. (2011). Importazole, a small molecule inhibitor of the transport receptor importin- β . *ACS Chemical Biology*, 6(7), 700-708. <https://doi.org/10.1021/cb2000296>
- Wang, J., Ramirez, T., Ji, P., Jayapal, S. R., Lodish, H. F., & Murata-Hori, M. (2012). Mammalian erythroblast enucleation requires PI3K-dependent cell polarization. *Journal of Cell Science*, 125(Pt 2), 340-349. <https://doi.org/10.1242/JCS.088286>
- Wölwer, C. B., Pase, L. B., Pearson, H. B., Gödde, N. J., Lackovic, K., Huang, D. C. S., Russell, S. M., & Humbert, P. O. (2015). A Chemical Screening Approach to Identify Novel Key Mediators of Erythroid Enucleation. *PLOS ONE*, 10(11), e0142655. <https://doi.org/10.1371/JOURNAL.PONE.0142655>
- Wölwer, C. B., Pase, L. B., Russell, S. M., & Humbert, P. O. (2016). Calcium signaling is required for erythroid enucleation. *PLOS ONE*, 11(1), 1-12. <https://doi.org/10.1371/journal.pone.0146201>
- Yoshino H, Sato Y, Nakano M. (2021). KPNB1 Inhibitor Importazole Reduces Ionizing Radiation-Increased Cell Surface PD-L1 Expression by Modulating Expression and Nuclear Import of IRF1. *Curr Issues Mol Biol.* 19;43(1):153-162. <https://doi.org/10.3390/cimb43010013>

Second decision letter

MS ID#: jcs.264385R1

MS Title: CDK9 interacts with a RanGTP-Importin- β complex to regulate erythroid enucleation

Authors: Lucas Newton; Krystle Lim; Christina Wölwer; Donia Abeid; Chad Johnson; Sarah Russell; Edwin Hawkins; Patrick Humbert

Article Type: Transfer

Dear Dr Newton,

I am happy to tell you that your manuscript has been accepted for publication in Journal of Cell Science, pending standard publication integrity checks.

Reviewer 2

Advance summary and potential significance to field

The authors have adequately addressed my comments.